# Unified Time Series Explanations via Amortized Optimization and Instance-level Multi-Expert Knowledge Distillation

**Viet-Hung Tran** [1]  **Zichi Zhang** [1]  **Ngoc Phu Doan** [1]  **Xuan Hoang Nguyen** [1]  **Phi Hung Nguyen** [1]  **Yimeng An** [1,2]
**Peixin Li** [1]  **Hans Vandierendonck** [1]  **Ira Assent** [3]  **Son Thai Mai** [1]

## Abstract

Deep neural networks (DNNs) achieve high accuracy on time series classification (TSC) but remain opaque, hindering deployment in sensitive domains. Existing post-hoc TSC explanation methods rely on a single attribution perspective and incur high per-instance computational cost, limiting real-time use. We propose XMA, a framework that unifies multiple post-hoc explainers at the instance level and amortizes the cost of explanation into a learned inference pass. *Instance-level Multi-Expert Knowledge Distillation (IMEKD)* selects, per instance, the attribution map with the highest combined faithfulness and robustness score from a pool of XAI experts. *Objective-Regularized Amortized Optimization Explanation (ORAOE)* trains an explainer DNN to reproduce this instance-best supervision while directly optimizing differentiable faithfulness and robustness objectives, letting the student refine beyond any individual teacher. *Faithfulness-Preserving Segmentation (FPS)* converts pointwise maps into contiguous segments without changing their faithfulness score. Across four synthetic datasets, MIT-ECG with cardiologist-verified ground truth, and 11 multivariate UEA benchmarks on three DNN architectures, XMA attains the best AUPRC on all synthetic and MIT-ECG benchmarks among compared methods, and the highest faithfulness on 7 of 11 and robustness on 8 of 11 UEA datasets, while amortizing explanation into a lightweight learned explainer rather than relying on per-instance optimization.

---

[1]The Queen's University Belfast, Belfast, Northern Ireland, United Kingdom [2]North China University of Water Resources and Electric Power, China [3]Aarhus University, Denmark. Correspondence to: Viet-Hung Tran <h.tran@qub.ac.uk>, Son Thai Mai <ThaiSon.Mai@qub.ac.uk>.

*Proceedings of the $43^{rd}$ International Conference on Machine Learning*, Seoul, South Korea. PMLR 306, 2026. Copyright 2026 by the author(s).

## 1. Introduction

Time series classification (TSC) is a fundamental task across diverse critical domains, ranging from finance (Liu & Cheng, 2025) to health care (Kampouraki et al., 2008; Jain et al., 2024). While DNNs have revolutionized natural language processing and computer vision, they have also emerged as a highly effective paradigm for TSC, particularly in modeling high-dimensional data with long-term temporal dependencies. For example, Transformer-based architectures such as PatchTST (Nie et al., 2023) have achieved strong performance on TSC benchmarks. However, the architectural complexity that drives this performance renders these models opaque to human users (Rudin, 2018). This lack of transparency undermines trust, complicates model auditing, and hinders the deployment of DNNs in safety-critical systems. XAI for time series has therefore emerged as a vital research area, bridging the gap between high-performance modeling and human interpretability.

XAI for TSC can be broadly categorized into two methodological perspectives. The first is *ante-hoc* explanation, which develops inherently interpretable architectures that provide explanations by design. The second is *post-hoc* explanation, which explains a fixed, pre-trained model without altering its training process or parameters. In both settings, a prevalent approach is the attribution map, which assigns each input feature an importance score proportional to its contribution to the classification outcome. Most methods assign these scores at the granularity of individual time points, in both ante-hoc (Lin & Runger, 2017; Hsu et al., 2019; Hosseini et al., 2020; Hsieh et al., 2021; Dang et al., 2020; Schwenke & Atzmueller, 2021) and post-hoc settings (Crabbé & Van Der Schaar, 2021; Queen et al., 2023; Liu et al., 2024a; Selvaraju et al., 2017). However, point-wise attributions often fail to align with human intuition, as they neglect the temporal dependencies and structural patterns inherent in time series. Recent research has therefore shifted towards subsequence-based explanations, which identify meaningful temporal segments rather than isolated points, in both ante-hoc designs (Wen et al., 2025; Senin & Malinchik, 2013; Nguyen et al., 2019) and post-hoc frameworks (Sivill & Flach, 2022; Tran et al., 2025b; Huang et al., 2025;

Tran et al., 2025a). Beyond feature attribution, alternative paradigms provide complementary insights. Instance-based explanations, such as counterfactuals, identify the minimal perturbation required to alter a model's prediction, in both ante-hoc (Ming et al., 2019; Ito & Chakraborty, 2019) and post-hoc contexts (Delaney et al., 2021). Symbolic approaches use logical rules, causality, or temporal logic to produce structured, semantic explanations of TSC models (Okajima & Sadamasa, 2019; Huang et al., 2020).

Among these methods, post-hoc explanation has become increasingly critical given the rapid adoption of complex but effective black-box DNNs. However, the combination of large model scale and the high dimensionality of multivariate time series imposes a prohibitive computational burden on traditional attribution methods. For example, Integrated Gradients-based methods (Tran et al., 2025b; Sundararajan et al., 2017) approximate a path integral that requires many queries to the target model $f_\theta$ for every test instance. This cost is not merely an inconvenience in safety-critical monitoring. For patients in ventricular tachycardia or fibrillation, the risk of death increases by roughly 5–10% per minute of delay before defibrillation, yet defibrillation is delayed beyond two minutes in about 30% of cases (Segall et al., 2015), and clinicians need inspectable explanations to act on advice (Taleban et al., 2025). Such settings also monitor many patients at once, so per-instance explanation cost accumulates (Nawaz & Ahmed, 2022). On 200 MIT-ECG instances, XMA produces explanations in 0.23 seconds versus 152.3 seconds for MIX (Appendix D.4), a margin that determines whether explanation-assisted monitoring is feasible where compute is limited. Algorithmic efficiency is therefore not a convenience but a prerequisite for deploying XAI in time- and resource-constrained environments.

Despite progress in post-hoc explanation, the current landscape remains fragmented. Existing methods exhibit substantial performance variability across individual instances, yet this heterogeneity presents an overlooked opportunity. No single explanation method is universally superior across all samples (Krishna et al., 2024; Han et al., 2022), a disagreement also documented for time series explainers (Nguyen et al., 2023). We observe empirically, under a combined faithfulness and robustness criterion, that for any given instance one "XAI expert" typically identifies the critical features best. Rather than committing to a single global method like existing works, we select and apply the best-performing expert for each instance.

**Contributions.** We introduce *XMA (eXplanation via Multi-expert and Amortized optimization)*, a framework that targets both the efficiency bottleneck and explanation quality in TSC XAI through knowledge distillation from multiple XAI experts and objective-regularized amortized optimization. Rather than querying the target model many times for

each test instance like existing works (Tran et al., 2025b; Sundararajan et al., 2017; Sivill & Flach, 2022), XMA shifts this computational burden to training: at inference, the default XMA performs one backward pass through the target model for gradient features and one forward pass of the explainer, while a gradient-free *Input-Only* variant removes the backward pass for fully forward-only inference. By combining instance-adaptive supervision with objective-regularized amortized training, in which the explainer directly optimizes faithfulness and robustness objectives, XMA produces attribution maps that exceed the quality of the individual experts. Our contributions are summarized below.

First, we introduce **Instance-level Multi-Expert Knowledge Distillation (IMEKD)**, which turns the instance-level inconsistency of post-hoc explainers into a source of supervision. Rather than committing to a single static method, IMEKD selects the most effective expert per instance and uses its attribution map to supervise the explanation DNN.

Second, we integrate IMEKD into **Objective-Regularized Amortized Optimization Explanation (ORAOE)**, which combines objective-regularized amortized optimization with multi-expert distillation for TSC XAI. Unlike standard distillation, which is bounded by teacher quality, ORAOE lets the explainer transcend imitation by directly optimizing differentiable faithfulness and robustness objectives, refining explanations beyond the individual experts.

Third, we propose **Faithfulness-Preserving Segmentation (FPS)**, which converts point-wise attribution maps into interpretable segments via an adaptive ratio step while provably preserving the faithfulness score. FPS bridges the efficiency of amortized attribution and the semantic clarity of segment-based explanations.

Fourth, XMA reduces inference to a learned explainer pass with a single gradient query to the target model, achieving substantially lower per-instance cost than optimization- and sampling-based explainers while matching the speed of simple gradient methods. We provide a formal complexity analysis and empirical runtime, FLOPs, parameter, and memory comparisons.

Finally, we evaluate XMA across four synthetic datasets with known motifs, the MIT-ECG dataset with cardiologist annotations, and 11 multivariate UEA datasets. XMA achieves the best AUPRC on all four synthetic datasets and on MIT-ECG, where it surpasses recent baselines including TIMING and Implet. On the 11 UEA datasets, XMA attains the highest faithfulness on 7 of 11 and robustness on 8 of 11, confirming the value of combining instance-adaptive supervision with objective-regularized amortized optimization.

**Conflict of Interest Disclosure.** The authors declare no financial conflicts of interest.

## 2. Background

### 2.1. Explanation for TSC DNNs

**Time series.** Let $\mathbf{x} = (x_1, \ldots, x_T) \in \mathbb{R}^{T \times d}$ denote a time series of length $T$ with $d$ variables (or channels), where each $x_t \in \mathbb{R}^d$ represents the feature vector at time step $t$ ($d = 1$ for univariate and $d > 1$ for multivariate time series) (Bagnall et al., 2016).

**Time series classification.** We denote the dataset as $\mathcal{D} = \{(\mathbf{x}^{(i)}, y^{(i)})\}_{i=1}^N$, where $N$ is the number of samples and $y^{(i)} \in \{1, \ldots, C\}$ is the ground-truth label for $C$ classes. The objective of TSC is to learn a mapping $F : \mathbb{R}^{T \times d} \to \{1, \ldots, C\}$ that predicts the class of a time series (Fawaz et al., 2018). In deep learning, this function is parameterized as a neural network $f_\theta : \mathbb{R}^{T \times d} \to \mathbb{R}^C$ with trainable parameters $\theta$, where the predicted class is obtained via $\hat{y} = \operatorname{argmax}(f_\theta(\mathbf{x}))$.

**Post-hoc explanation for deep learning in TSC.** Given a trained TSC model $f_\theta : \mathbb{R}^{T \times d} \to \mathbb{R}^C$, post-hoc explanation aims to explain the decision-making process of $f_\theta$ for an input $\mathbf{x}$ without altering the model structure or its parameters $\theta$. A common approach is the *attribution map*, which assigns an importance score to each feature of the time series, where a higher score indicates a greater contribution to the model's prediction. The notion of a "feature" here can vary: it may refer to individual time steps $x_t$, segments $\mathbf{x}_{i:j} = (x_i, \ldots, x_j)$ with $1 \le i < j \le T$, or specific channels. In general, we define an attribution map as a sequence $\mathbf{A} = (s_1, s_2, \ldots, s_T) \in \mathbb{R}^{T \times d}$, where each $s_t \in \mathbb{R}^d$ is a vector of importance scores for the $d$ variables at time step $t$.

### 2.2. Amortized Optimization

Amortized Optimization is a paradigm that utilizes machine learning to efficiently solve repeated instances of an optimization problem. Formally, for a given context $x$, we seek to solve the objective:

$$y^*(x) \in \operatorname*{argmin}_{y \in Y} f(x, y). \tag{1}$$

**Definition 2.1** (Amos et al. (2023))**.** An amortized optimization method for solving (1) is defined as the tuple $\mathcal{A} := (f, Y, X, p, \hat{y}_\theta, \mathcal{L})$. In this formulation, $X$ and $Y$ denote the context space and solution domain, respectively, while $f : X \times Y \to \mathbb{R}$ represents the unconstrained objective function. The term $p$ denotes the probability distribution over the contexts $X$. Finally, $\hat{y}_\theta : X \to Y$ is an amortization model parameterized by $\theta$, which is determined by minimizing a loss function $\mathcal{L}(f, Y, X, p, \hat{y}_\theta)$ defined over the tuple components.

Amortized optimization methods are generally categorized into two classes based on their interaction with the objective function during learning or inference:

**Definition 2.2** (Amos et al. (2023))**.** A *fully-amortized model* $\hat{y}_\theta : X \to Y$ maps the context directly to a solution of (1) without querying the objective function $f$.

**Definition 2.3** (Amos et al. (2023))**.** A *semi-amortized model* $\hat{y}_\theta : X \to Y$ maps the context to a solution while incorporating evaluations of the objective $f$ defined in (1), typically via an iterative refinement procedure.

XMA is a fully-amortized model augmented with objective regularization: the faithfulness and robustness objectives shape training only, and the inference remains a single forward pass.

## 3. Our proposed XMA framework

This section presents **XMA** (*eXplanation via Multi-expert and Amortized optimization*), our framework for post-hoc TSC explanation. As shown in Figure 1, XMA proceeds in two stages: Instance-level Multi-Expert Knowledge Distillation (IMEKD) and Objective-Regularized Amortized Optimization Explanation (ORAOE). In IMEKD (Section 3.1), for each instance we generate several attribution maps from a pool of experts and select the one with the highest combined faithfulness and robustness score, where these criteria are defined in Sections 3.2 and 4.1. Each resulting (instance, attribution map) pair becomes training data for ORAOE. In ORAOE (Section 3.2), we train an amortized explainer DNN that maps an instance and its $\text{Input} \times \text{Gradient}$ features to the selected attribution map, then refines that map beyond the IMEKD selection by directly optimizing the faithfulness and robustness objectives. Finally, we apply FPS (Section 3.3) to the resulting map, assigning each segment the average score of its constituent time points.

### 3.1. Instance-level Multi-Expert Knowledge Distillation

**Explanation experts.** Given a dataset $\mathcal{D}$ and a trained model $f_\theta$, we define an explanation method as a function $\mathcal{E} : (\mathbf{x}, f_\theta) \to \mathbf{A}$ that produces an attribution map $\mathbf{A} \in \mathbb{R}^{T \times d}$ for an instance $\mathbf{x}$. We refer to each such method $\mathcal{E}$ (e.g., Integrated Gradients (Sundararajan et al., 2017), LIME (Ribeiro et al., 2016), or KernelSHAP (Lundberg & Lee, 2017)) as an *XAI expert*.

**Instance-level best expert.** Let $\mathbb{E} = \{\mathcal{E}_1, \ldots, \mathcal{E}_K\}$ denote a pool of $K$ candidate experts. We define a selection score $\mathcal{S}(\mathbf{x}, f_\theta, \mathbf{A}) \in \mathbb{R}$ that quantifies the quality of an attribution map $\mathbf{A}$ for instance $\mathbf{x}$ under model $f_\theta$, where higher values indicate better explanations. For each time series $\mathbf{x}^{(i)} \in \mathcal{D}$, we select the *instance-best expert* $\mathcal{E}_{\mathbf{x}^{(i)}}^*$ as the one whose attribution map maximizes this score:

$$\mathcal{E}_{\mathbf{x}^{(i)}}^* = \operatorname*{argmax}_{\mathcal{E} \in \mathbb{E}} \mathcal{S}\big(\mathbf{x}^{(i)}, f_\theta, \mathcal{E}(\mathbf{x}^{(i)}, f_\theta)\big). \tag{2}$$

We instantiate $\mathcal{S}$ as the sum of the negated faithfulness loss ($-\mathcal{L}_{\text{faith}}$; Eq. 6) and the Jaccard-based robustness score over

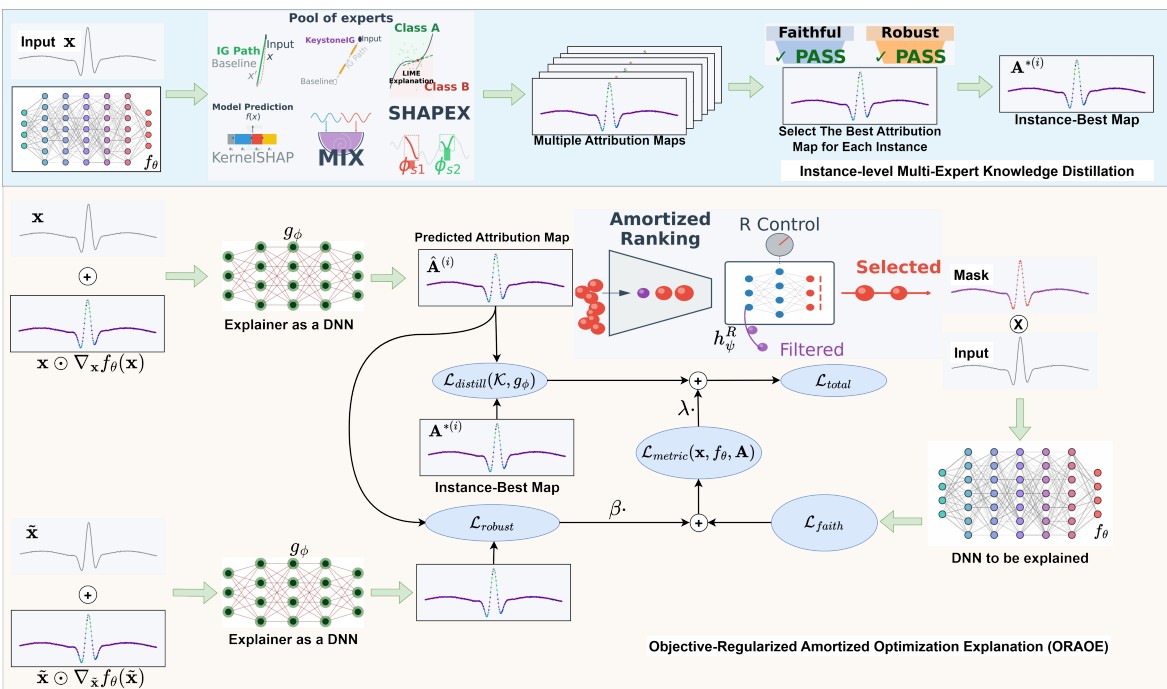

Figure 1. **Overview of the XMA framework.** XMA consists of two stages. **(Top) Instance-level Multi-Expert Knowledge Distillation (IMEKD).** For each input instance, we generate a diverse set of attribution maps using a pool of experts (e.g., LIME, KernelSHAP, IG) and select the instance-best map $\mathbf{A}^*$ under a combined faithfulness and robustness score (green checks). **(Bottom) Objective-Regularized Amortized Optimization Explanation (ORAOE).** We train a parameterized explainer $g_\phi$ to predict $\mathbf{A}^*$ under the total objective $\mathcal{L}_{\text{total}} = \mathcal{L}_{\text{distill}} + \lambda \mathcal{L}_{\text{metric}}$, combining a *distillation loss* ($\mathcal{L}_{\text{distill}}$, MSE to the selected expert map) with a *metric loss* ($\mathcal{L}_{\text{metric}} = \mathcal{L}_{\text{faith}} + \beta \mathcal{L}_{\text{robust}}$). The *faithfulness loss* $\mathcal{L}_{\text{faith}}$ uses a proxy ranking gate to maintain model confidence on masked inputs, and the *robustness loss* $\mathcal{L}_{\text{robust}}$ minimizes attribution discrepancy under input noise. The explainer is conditioned on the raw input concatenated with its $\text{Input} \times \text{Grad}$ map.

the top-20% salient features (Section 4.1), both evaluated per instance. Because expert selection requires no gradient, $\mathcal{S}$ uses the exact, non-differentiable forms: the hard-mask faithfulness loss of Eq. 6 and the discrete Jaccard robustness metric. In contrast, training the explainer (Section 3.2) requires differentiable surrogates: the gate-approximated faithfulness loss (Eq. 9) and the map-discrepancy robustness loss $\mathcal{L}_{\text{robust}}$ (Eq. 10).

**Pool construction.** We construct the pool $\mathbb{E}$ to maximize methodological diversity, combining perturbation-based experts (e.g., LIME, KernelSHAP) with gradient-based experts (e.g., Integrated Gradients, Input×Gradient). For each instance, every expert in the pool produces an attribution map; we score each map with the selection score $\mathcal{S}$, and we take the highest-scoring map as that instance's teacher. We emphasize that $\mathbf{A}^{*(i)}$ is the best map *among the experts in the pool* for instance $\mathbf{x}^{(i)}$, not a global optimum over all possible attribution maps; the amortized training in Section 3.2 can refine the explanation beyond this pool-restricted teacher. A small, diverse pool suffices: pool-size and expert-removal analyses (Appendix C.6) show that most of the benefit comes from the first few experts and that performance degrades gracefully when an expert is removed.

The specific pool composition for each benchmark is given in Appendix A below.

**Multi-expert knowledge.** We define the *multi-expert knowledge* set $\mathcal{K} = \{(\mathbf{x}^{(i)}, \mathbf{A}^{*(i)})\}_{i=1}^N$, where $\mathbf{A}^{*(i)} = \mathcal{E}^*_{\mathbf{x}^{(i)}}(\mathbf{x}^{(i)}, f_\theta)$ is the attribution map produced by the instance-best expert for $\mathbf{x}^{(i)}$.

### 3.2. Objective-Regularized Amortized Optimization Explanation

**Objective.** We cast explanation as an optimization problem: find the attribution map $\mathbf{A}$ that maximizes an evaluation metric $\mathcal{M}$ (faithfulness, robustness, or a combination). For an instance $\mathbf{x}$ and model $f_\theta$, the conceptual optimum $\mathbf{A}^{\text{opt}}$ is defined as:

$$\mathbf{A}^{\text{opt}} = \underset{\mathbf{A} \in \mathbb{R}^{T \times d}}{\arg\max} \mathcal{M}(\mathbf{x}, f_\theta, \mathbf{A}). \tag{3}$$

Computing $\mathbf{A}^{\text{opt}}$ exactly for every instance is intractable; the amortized explainer instead learns to approximate it under supervision from the expert-selected maps of Section 3.1. For each training instance $\mathbf{x}^{(i)}$, the explainer is supervised by a single teacher map $\mathbf{A}^{*(i)}$, the instance-best expert output selected by IMEKD (Eq. 2); non-selected expert maps

are not used.

**Amortized Optimization Setup.** Solving Eq. 3 directly for every instance via iterative optimization is computationally expensive. We instead adopt an *amortized optimization* framework, treating the pair $(\mathbf{x}, f_\theta)$ as the *context* and $\mathbf{A}^{\mathrm{opt}}$ as the *solution*. Rather than optimizing $\mathbf{A}$ from scratch per instance, we learn a parameterized explainer $g_\phi$ with parameters $\phi$. To make the explainer sensitive to $f_\theta$, we condition it on both the input and its local gradient by concatenating $\mathbf{x}$ with the *Input × Gradient* map (Shrikumar et al., 2016), $\mathbf{I}_{\mathrm{grad}} = \mathbf{x} \odot \nabla_{\mathbf{x}} f_\theta(\mathbf{x})$. The explainer is thus a mapping $g_\phi : \mathbb{R}^{T \times 2d} \to \mathbb{R}^{T \times d}$ with prediction $\hat{\mathbf{A}} = g_\phi(\mathrm{Concat}(\mathbf{x}, \mathbf{I}_{\mathrm{grad}}))$. The gradient supplies the model's first-order local sensitivity around $\mathbf{x}$, which helps the explainer adapt to the decision boundary of $f_\theta$. Computing it requires one backward pass through $f_\theta$ per instance, so the default XMA performs one forward and one backward pass on the target model followed by a single forward pass of the explainer. The gradient-free *Input-Only* variant (Section 4.4) omits the backward pass and is therefore fully forward-only, trading a small amount of faithfulness for the lowest inference cost.

**Objective-Regularized Amortized Optimization.** To train $g_\phi$, we combine multi-expert distillation with direct metric optimization in a composite objective $\mathcal{L}_{\mathrm{total}}$:

$$\mathcal{L}_{\mathrm{total}} = \mathcal{L}_{\mathrm{distill}}(\mathcal{K}, g_\phi) + \lambda \cdot \mathcal{L}_{\mathrm{metric}}(\mathbf{x}, f_\theta, \mathbf{A}), \quad (4)$$

where $\lambda \in [0, 1]$ balances the two terms. The distillation term $\mathcal{L}_{\mathrm{distill}}$ guides the student $g_\phi$ toward the expert-selected maps $\mathbf{A}^{*(i)}$ in $\mathcal{K}$. The metric term $\mathcal{L}_{\mathrm{metric}}$ optimizes faithfulness and robustness directly, letting the explainer refine beyond the fixed experts rather than merely imitate them. $\mathcal{L}_{\mathrm{metric}}$ is a differentiable training surrogate for the evaluation metric $\mathcal{M}$: it shares the faithfulness term ($\mathcal{L}_{\mathrm{faith}}$), but replaces the non-differentiable Jaccard robustness used in $\mathcal{M}$ with the differentiable map-discrepancy loss $\mathcal{L}_{\mathrm{robust}}$ (Eq. 10) so that the objective is trainable by gradient descent. Because this evaluation-derived objective regularizes the amortized explainer, we term the method *objective-regularized amortized optimization*.

**Distillation Loss.** The distillation loss is the mean squared error between the predicted map $\hat{\mathbf{A}}^{(i)} = g_\phi(\mathrm{Concat}(\mathbf{x}^{(i)}, \mathbf{I}_{\mathrm{grad}}^{(i)}))$ and the expert-selected map $\mathbf{A}^{*(i)}$ from $\mathcal{K}$:

$$\mathcal{L}_{\mathrm{distill}}(\mathcal{K}, g_\phi) = \frac{1}{N} \sum_{i=1}^{N} \left\| \hat{\mathbf{A}}^{(i)} - \mathbf{A}^{*(i)} \right\|_F^2, \quad (5)$$

where $\| \cdot \|_F$ denotes the Frobenius norm. Because expert maps from different methods differ in scale and sign convention, we apply per-instance min-max normalization to all attribution maps before computing the MSE, aligning them to a common range.

**Faithfulness Loss.** We adapt the masking-based faithfulness evaluation proxy of (Queen et al., 2023; Liu et al., 2024a) into a differentiable training objective: if $\mathbf{A}$ identifies the most salient features, retaining only the top features while masking the rest should be sufficient for the model to maintain high confidence in the target class. Let $\mathcal{R}_\chi = \{ i \cdot \chi \mid 1 \le i \le \lfloor 1/\chi \rfloor \}$ be a set of masking ratios with step size $\chi$ (default $\chi = 0.1$). For each $R \in \mathcal{R}_\chi$, let $\mathbf{m}_R \in \{0, 1\}^{T \times d}$ be a binary mask in which 1 marks the top $(1 - R)$ features of $\mathbf{A}$. The ideal faithfulness loss is

$$\mathcal{L}_{\mathrm{faith}} = \sum_{R \in \mathcal{R}_\chi} -\log \Big( f_\theta \big( \mathbf{m}_R \odot \mathbf{x} \\ + (1 - \mathbf{m}_R) \odot \mathbf{x}' \big)_{y_{\mathrm{true}}} \Big), \quad (6)$$

where $y_{\mathrm{true}}$ is the true label index of $\mathbf{x}$ and $\mathbf{x}'$ is a baseline. We use the per-feature training mean as the baseline $\mathbf{x}'$, which is approximately zero under z-score normalization; we report sensitivity to this choice in Appendix C.3.

**Differentiable Mask Approximation.** Constructing $\mathbf{m}_R$ requires non-differentiable sorting. We therefore train a secondary network, the *Faithfulness Gate* $h_\psi^R : \mathbb{R}^{T \times d} \to [0, 1]^{T \times d}$ with parameters $\psi$, to approximate the hard masking $\hat{\mathbf{m}}_R = h_\psi^R(\mathbf{A})$. The gate is trained to mimic the sorting logic by minimizing the binary cross-entropy between its output and the hard mask $\mathbf{m}_R = \mathrm{HardMask}_R(\mathbf{A})$, over a distribution of random attribution maps:

$$\psi^* = \underset{\psi}{\arg\min} \, \mathbb{E}_{\mathbf{A}} \left[ \mathcal{L}_{\mathrm{BCE}} \big( h_\psi^R(\mathbf{A}), \mathrm{HardMask}_R(\mathbf{A}) \big) \right], \quad (7)$$

where the element-wise binary cross-entropy is

$$\mathcal{L}_{\mathrm{BCE}}(\hat{\mathbf{y}}, \mathbf{y}) = -\frac{1}{T \cdot d} \sum_k \Big( y_k \log \hat{y}_k \\ + (1 - y_k) \log(1 - \hat{y}_k) \Big), \quad (8)$$

with $\mathbf{y}$ the binary mask, $\hat{\mathbf{y}}$ the predicted mask, and $y_k, \hat{y}_k$ their values at position $k$. The gate is trained on random attribution maps but applied to real maps produced by $g_\phi$. Because all attribution maps are per-instance min-max normalized to $[0, 1]$ (as introduced for the distillation loss above), the gate operates on a common rank-on-$[0, 1]$ representation at both training and inference time, eliminating magnitude-level distribution shift between the two regimes. Substituting this differentiable proxy yields the final faithfulness loss as follows:

$$\mathcal{L}_{\mathrm{faith}} = \frac{1}{N} \sum_{i=1}^{N} \sum_{R \in \mathcal{R}_\chi} -\log \Big( f_\theta \big( h_\psi^R(\hat{\mathbf{A}}^{(i)}) \odot \mathbf{x}^{(i)} \\ + (1 - h_\psi^R(\hat{\mathbf{A}}^{(i)})) \odot \mathbf{x}' \big)_{y_{\mathrm{true}}} \Big). \quad (9)$$

**Robustness Loss.** The robustness loss stabilizes the explainer by minimizing the discrepancy between attribution maps from the original input $\mathbf{x}$ and a perturbed version $\tilde{\mathbf{x}} = \mathbf{x} + \boldsymbol{\epsilon}$, with $\boldsymbol{\epsilon} \sim \mathcal{N}(0, \sigma^2 \mathbf{I})$. This stability objective is related to the relative input and output stability metrics used for evaluating explanations (Agarwal et al., 2022), which we adapt here as a differentiable training loss. The perturbed input-gradient feature is $\tilde{\mathbf{I}}_{\text{grad}} = \tilde{\mathbf{x}} \odot \nabla_{\tilde{\mathbf{x}}} f_\theta(\tilde{\mathbf{x}})$, and the loss is the expected squared difference between the two explanations:

$$\mathcal{L}_{\text{robust}} = \frac{1}{N} \sum_{i=1}^{N} \left\| g_\phi([\mathbf{x}^{(i)}; \mathbf{I}_{\text{grad}}^{(i)}]) - g_\phi([\tilde{\mathbf{x}}^{(i)}; \tilde{\mathbf{I}}_{\text{grad}}^{(i)}]) \right\|_F^2. \tag{10}$$

**Metric Loss.** We aggregate faithfulness and robustness into the metric loss that drives the objective-regularized term of $\mathcal{L}_{\text{total}}$ as follows:

$$\begin{aligned}\mathcal{L}_{\text{metric}}(\mathbf{x}, f_\theta, \mathbf{A}) =\ &\mathcal{L}_{\text{faith}}(\mathbf{x}, f_\theta, \mathbf{A}) \\ &+ \beta \cdot \mathcal{L}_{\text{robust}}(\mathbf{x}, f_\theta, \mathbf{A}),\end{aligned} \tag{11}$$

where $\beta$ controls the faithfulness–robustness trade-off.

**Choice of target label.** We define $\mathcal{L}_{\text{faith}}$ with respect to the true label $y_{\text{true}}$; both the true and predicted labels are used as explanation targets in the broader XAI literature (Gat et al., 2022). On correctly predicted instances, where $y_{\text{true}} = y_{\text{pred}}$, the choice of $y_{\text{true}}$ versus $y_{\text{pred}}$ is immaterial. On misclassified instances the two targets diverge, and an explanation faithful to $y_{\text{true}}$ need not be faithful to the model's actual decision. We analyze this case empirically in Appendix C.1.

### 3.3. Faithfulness-Preserving Segmentation

To align the fine-grained attribution map $\mathbf{A}$ with human visual perception, we introduce a post-processing module that aggregates individual time-step scores into contiguous segments. This aggregation is designed to leave the faithfulness metric invariant at a specified granularity step $\delta$.

**Faithfulness Metric Definition.** We measure faithfulness by the *Area Under the Performance Curve* (AUPC) under feature removal. Let $\mathcal{P}(f_\theta, \mathcal{D}, R)$ denote the ROC-AUC of $f_\theta$ on $\mathcal{D}$ after masking the least important $R$ fraction of features (equivalently, retaining the top $1 - R$), as in our evaluation protocol (Section 4.1). The faithfulness score integrates this performance over the sparsity range $\mathcal{R}_\delta = \{k \cdot \delta \mid 0 \leq k \leq \lceil 1/\delta \rceil\}$ with step size $\delta$:

$$\text{Faithfulness} = \sum_{k=0}^{\lceil 1/\delta \rceil} \mathcal{P}(f_\theta, \mathcal{D}, \min(1, k \cdot \delta)) \cdot \delta. \tag{12}$$

The $k = 0$ term corresponds to the unmasked model and contributes a method-independent constant, so it does not affect the relative ranking of explanation methods.

**Segmentation Algorithm.** To produce segments without altering this score, we apply a *stratified merging* strategy. We partition features into importance bins defined by the cumulative ratios of $\delta$ (e.g., $0 \rightarrow \delta, \delta \rightarrow 2\delta, 2\delta \rightarrow 3\delta$). For the bin spanning the rank interval $[R - \delta, R]$ with $R \in \mathcal{R}_\delta$, we identify all time steps in that rank range and merge adjacent steps into contiguous segments. We then assign each segment a uniform score equal to the average importance of its constituent time steps.

**Faithfulness-Preserving Guarantee.** Because averaging occurs only *within* a single rank bin and never across bins, the ranking of features at resolution $\delta$ is unchanged. The set of features removed at any threshold $k \cdot \delta$ is therefore identical for the original and segmented maps, so the faithfulness score of Eq. 12 is provably preserved. We provide the formal proof in Appendix E.

## 4. Experiments

### 4.1. Experimental Setup

**Datasets and Models.** We evaluate our framework across three distinct domains. First, following the SHAPEX protocol (Huang et al., 2025), we use four synthetic datasets (MCCE, MCCH, MTCE, MTCH) with a Transformer backbone (Queen et al., 2023). Second, for real-world validation, we utilize the annotated MIT-ECG dataset (Queen et al., 2023) with a CNN architecture. Third, we extend to the multivariate setting using 11 datasets from the UEA archive (with abbreviations in parentheses): UWaveGestureLibrary (UWave), ERing, RacketSports (Racket), NATOPS, CharacterTrajectories (Character), SelfRegulationSCP1 (SR SCP1), ArticularyWordRecognition (Articulary), Libras, BasicMotions (Motions), Cricket, Epilepsy. We employ three distinct architectures: ResNet, Fully-Convolutional Networks (FCN), and PatchTST, configured as in (Wen et al., 2025). Detailed statistics and preprocessing steps are provided in Appendix A.

**Baselines.** We compare against attribution methods suited to each setting. For the synthetic benchmarks, we evaluate against Integrated Gradients (Sundararajan et al., 2017), Dynamask (Crabbé & Van Der Schaar, 2021), TimeX (Queen et al., 2023), TimeX++ (Liu et al., 2024a), and SHAPEX (Huang et al., 2025). For MIT-ECG, we additionally include the recent MIX method (Tran et al., 2025a). The synthetic-specific baselines (TimeX, TimeX++, SHAPEX) are designed for univariate, motif-based data and do not transfer directly to the multivariate UEA setting; for UEA we therefore use multivariate-capable, general-purpose explainers: LIME (Ribeiro et al., 2016), KernelSHAP (Lundberg & Lee, 2017), Input×Gradient (Shrikumar et al., 2016), and Integrated Gradients, and additionally compare against the recent TSC explainers TIMING (Jang et al., 2025) and Implet (Meng et al., 2025) (Appendix B.3). Implementation

details are available in Appendix A .

**Evaluation Metrics.** We adopt a comprehensive evaluation protocol tailored to the availability of ground-truth explanations. *1) Ground-Truth Evaluation (Synthetic & MIT-ECG):* For datasets with known salient features, we treat attribution as a binary classification task at the time-step level. We report **Area Under the Precision-Recall Curve (AUPRC)**, **Area Under Precision (AUP)**, and **Area Under Recall (AUR)**, where higher scores indicate better alignment with ground-truth features. *2) Unsupervised Evaluation (UEA Multivariate):* For real-world data without ground truth, we rely on proxy metrics. First, for **Faithfulness**, we measure the AUC of the model's performance (ROC-AUC) as we sequentially mask the least important features at increasing sparsity ratios $r \in [0, 1]$; for multi-class UEA datasets, ROC-AUC is computed as macro-averaged one-vs-rest. A higher value implies the model retains accuracy despite feature removal (Huang et al., 2025; Queen et al., 2023; Liu et al., 2024a). Second, for **Robustness**, we evaluate stability under local perturbations (Tran et al., 2025a). We binarize attribution maps (top 20% salient features) and calculate the **Jaccard Index** between the salient sets of original and noisy inputs; higher overlap indicates greater resistance to noise.

### 4.2. Main Results

The comprehensive evaluation results for our XMA framework are presented in Figure 2.

**Synthetic & Real-World Ground Truth.** On the four synthetic benchmarks, XMA demonstrates superior precision, outperforming all baselines in AUPRC while achieving the top rank in AUP on 3 out of 4 datasets and AUR on 2 out of 4. This trend holds for the real-world MIT-ECG dataset, where our XMA technique surpasses the MIX method across all reported metrics. The MIT-ECG ground truth consists of cardiologist-verified annotations from the TimeX subset (Queen et al., 2023), giving expert-grounded validation that complements the proxy faithfulness and robustness metrics used on the UEA benchmarks.

**Multivariate Time Series.** On the 11 UEA datasets (where ground truth is unavailable), XMA exhibits strong generalization. It achieves the highest Faithfulness scores in 7 out of 11 datasets and the highest Robustness scores in 8 out of 11 datasets. Crucially, these results report the mean and standard deviation aggregated across three distinct DNN architectures. The low variance observed in the error bars highlights the consistency of XMA across different backbones, confirming its ability to generate explanations that are both reliable and stable under perturbation.

Additional qualitative visualizations and a detailed analysis are provided in Appendix B.

*Table 1.* Reverse ablation on MIT-ECG and MCCE. Removing distillation (Metric-Only) collapses AUPRC, while removing the metric loss (XM) is competitive but below the full model. Best per dataset in bold.

| Dataset | Variant | AUPRC |
|---------|---------|-------|
| MIT-ECG | Full XMA | **0.890** |
|         | XM (distillation only) | 0.875 |
|         | Metric-Only (no distillation) | 0.429 |
| MCCE    | Full XMA | **0.889** |
|         | XM (distillation only) | 0.773 |
|         | Metric-Only (no distillation) | 0.322 |

### 4.3. Computational Efficiency

As illustrated in Figure 3, XMA demonstrates superior computational efficiency compared to state-of-the-art baselines. It achieves orders-of-magnitude acceleration over optimization-based methods (SHAPEX, MIX) and sampling-heavy approaches (LIME, KernelSHAP), effectively bridging the gap between high-fidelity explanations and real-time inference. Notably, XMA matches the speed of simple gradient-based methods (e.g., Integrated Gradients), empirically corroborating the theoretical complexity analysis provided in Appendix D. XMA's FLOPs advantage scales with target-model complexity, from $19\times$ fewer FLOPs than IG on ResNet (209K parameters) to $47\times$ on PatchTST (6.39M parameters); a full breakdown of FLOPs, parameters, and memory is in Appendix D.3.

### 4.4. Ablation Study

We conduct an ablation study to evaluate the contribution of each component of XMA. On the UEA archive (Figure 4) we compare the full model against two variants: (1) **XM**, which removes the metric loss $\mathcal{L}_{\text{metric}}$ (Eq. 11) and trains on distillation alone, and (2) **Input-Only XMA**, which uses only the raw input without the target model's gradient (Input$\times$Gradient). On MIT-ECG and the MCCE synthetic dataset, we additionally include a **Metric-Only** variant that removes distillation entirely and trains on the metric loss alone (Table 1), isolating the role of expert distillation.

**Necessity of distillation.** Removing distillation collapses AUPRC, from 0.890 to 0.429 on MIT-ECG and from 0.889 to 0.322 on MCCE. Without a teacher, the explainer must discover attributions from the faithfulness and robustness gradients alone, a weak signal with many poor local minima. Distillation provides the warm start that makes the metric loss effective.

**Impact of the metric loss.** XM (distillation only) is competitive but cannot exceed the full model: on the UEA archive, full XMA yields higher faithfulness on 8 of 11 datasets and higher robustness across all tasks (Figure 4), and on MIT-ECG and MCCE it improves AUPRC over XM by 0.015

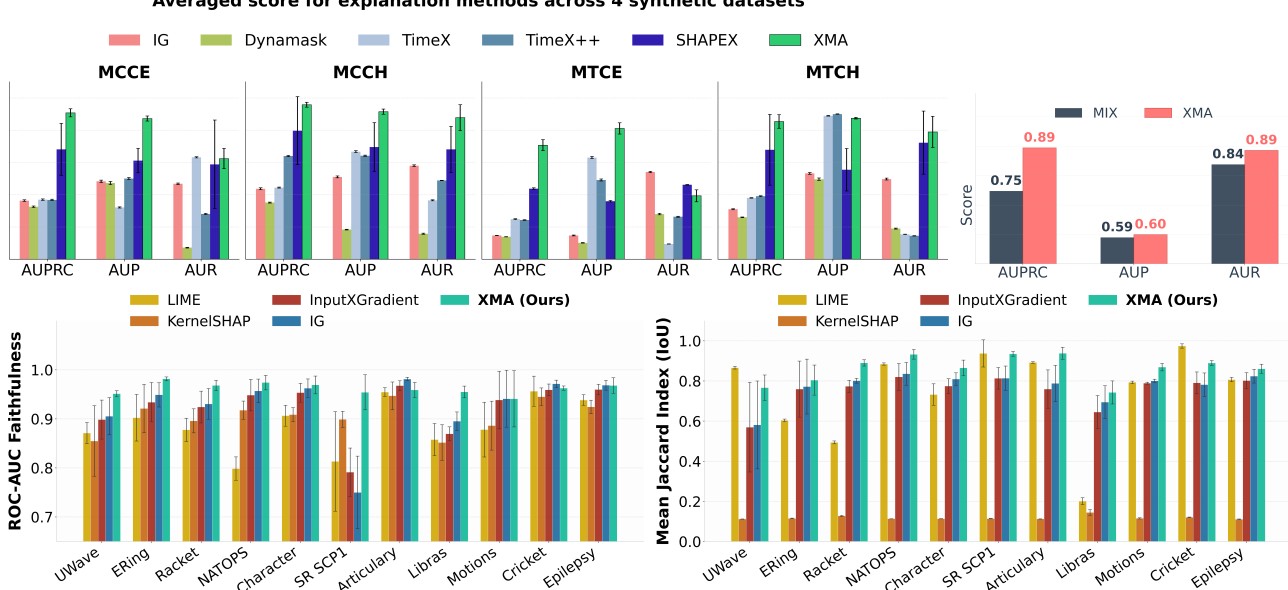

Figure 2. **Comprehensive evaluation of the XMA framework across synthetic and real-world domains. (Top Left)** Performance on four synthetic benchmarks (MCCE, MCCH, MTCE, MTCH). XMA (green) consistently outperforms five state-of-the-art baselines (IG, Dynamask, TimeX, TimeX++, SHAPEX) in AUPRC and AUP metrics. **(Top Right)** Validation on the MIT-ECG dataset. XMA (red) achieves superior anomaly localization compared to MIX, improving AUPRC by $\sim 0.14$. **(Bottom)** Faithfulness evaluation (ROC-AUC) on 11 multivariate UEA datasets. XMA (teal) attains the highest fidelity in **7 out of 11** tasks. Robustness analysis (Mean Jaccard Index). XMA demonstrates the highest stability under local perturbations in **8 out of 11** datasets, significantly outperforming baselines.

and 0.116, respectively. The two components are therefore complementary: distillation supplies a high-quality warm start, while the metric loss refines the explanation beyond teacher quality. This complementarity is the practical pay-off of the framework: a pool of general-purpose explainers (LIME, KernelSHAP, Integrated Gradients) supervises an amortized student that surpasses TSC-specialized methods (TIMING, Implet) on MIT-ECG AUPRC and on Libras faithfulness across three backbones (Appendix B.3).

**Architectural flexibility.** While the gradient input (Full XMA) maximizes faithfulness across all datasets, the **Input-Only XMA** variant (Figure 4) achieves higher robustness on 10 of 11 datasets despite lacking access to model internals. This allows deployment in black-box or resource-constrained settings where gradient queries are prohibitive, while still delivering stable and competitive explanations.

## 5. Related Work

Explainable AI (XAI) for TSC divides into *ante-hoc* and *post-hoc* approaches.

**Feature attribution** is the dominant strategy across both settings, assigning each input feature an importance score for its contribution to the prediction. Most methods operate at the granularity of individual time points, using gradient or perturbation techniques in both ante-hoc (Lin & Runger,

2017; Hsu et al., 2019; Hosseini et al., 2020; Hsieh et al., 2021; Dang et al., 2020; Schwenke & Atzmueller, 2021) and post-hoc settings (Crabbé & Van Der Schaar, 2021; Queen et al., 2023; Liu et al., 2024a; Sundararajan et al., 2017; Vinayavekhin et al., 2018; Karim et al., 2017), but point-wise attributions often miss the temporal dependencies inherent in time series. Recent work has therefore shifted toward **subsequence-based explanations** that identify contiguous segments rather than isolated points, in both interpretable architectures (Wen et al., 2025; Senin & Malinchik, 2013; Nguyen et al., 2019) and post-hoc frameworks (Sivill & Flach, 2022; Tran et al., 2025b; Cho et al., 2021; Huang et al., 2025; Tran et al., 2025a); the closest works on univariate time series are MIX (Tran et al., 2025a) and SHAPEX (Huang et al., 2025).

Several recent methods improve TSC attribution from a single methodological perspective: TIMING (Jang et al., 2025) augments Integrated Gradients with temporality-aware stochastic baselines and segment-based masking, ORTE (Yue et al., 2025) learns a binary mask via an adaptive generator with a straight-through estimator and a contrastive objective, and Implet (Meng et al., 2025) produces explanations as importance-weighted subsequences. XMA is orthogonal: rather than proposing another attribution method, it combines heterogeneous explainers in an instance-adaptive manner, and any of these can serve as an expert in its pool, including stronger explainers as

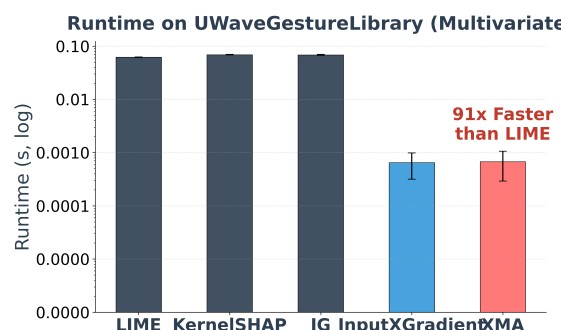

*Figure 3.* **Computational efficiency analysis.** Per-instance inference time (seconds, log scale) on **(Left)** univariate benchmarks (Synthetic and MIT-ECG) and **(Right)** multivariate UEA datasets. **XMA 1** and **XMA 64** denote our explainer running with batch sizes of 1 and 64. XMA benefits from batching because it produces an explanation in a single amortized forward pass, which processes many instances in parallel. The baselines cannot: IG requires roughly 50 sequential model queries per instance along its integration path, and LIME and KernelSHAP require roughly 25 perturbed-sample queries per instance, so their cost scales with the number of instances regardless of batching.

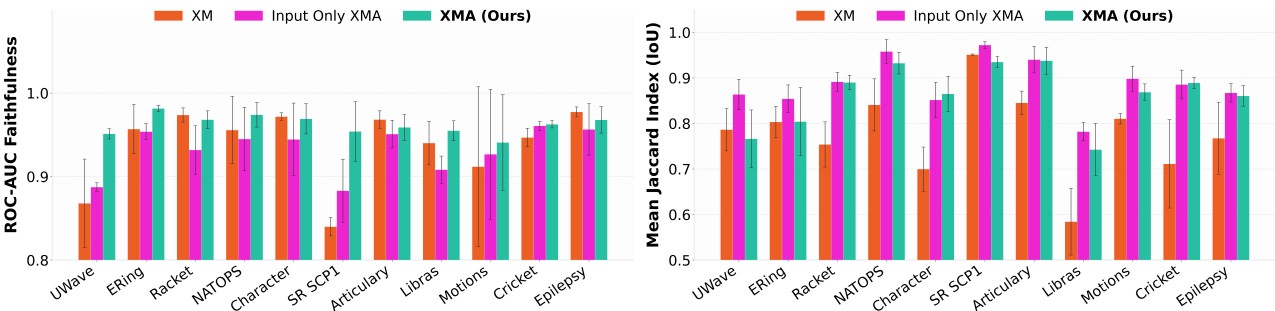

*Figure 4.* **Multivariate ablation study.** The full XMA framework (teal) generally yields superior faithfulness (left) and competitive robustness (right) relative to its ablated variants across UEA datasets.

they emerge. AMEE (Nguyen et al., 2023) shares our observation that explainers disagree, but it recommends one explainer *per dataset* and acts as an evaluator; XMA selects *per instance* and directly generates the explanation through a single amortized forward pass.

Beyond attribution, alternative paradigms offer complementary insights. **Instance-based explanations**, such as counterfactuals, characterize decision boundaries by identifying the minimal perturbation required to alter a model's prediction (Ming et al., 2019; Ito & Chakraborty, 2019; Delaney et al., 2021). **Symbolic reasoning** approaches use logical rules and causality to provide structured, semantic descriptions of model behavior (Okajima & Sadamasa, 2019; Huang et al., 2020; Mohammadinejad et al., 2020).

Two precedents most closely intersect XMA's design space. MIX (Tran et al., 2025a) selects among time-frequency views of a single input via iterative Integrated Gradients; XMA instead selects among heterogeneous XAI methods and amortizes inference into one learned forward pass with a single gradient query. Dynamask (Crabbé & Van Der Schaar, 2021) fits a perturbation mask per instance via iterative

optimization; XMA's amortized explainer replaces this per-instance solver with a network that produces the map in one forward pass. Empirically, XMA explains 200 MIT-ECG instances in 0.23s versus 152.3s for MIX (Appendix D.4).

## 6. Conclusion

In this paper, we introduce a framework for enhancing the interpretability of Deep Neural Networks in Time Series Classification. By leveraging *multi-expert knowledge distillation* at the instance level combined with *objective-regularized amortized optimization*, our approach effectively bridges the gap between explanation fidelity and computational efficiency. XMA unifies diverse attribution methods through instance-level selection while simultaneously addressing the bottleneck of inference latency, and, to the best of our knowledge, is among the first to bring objective-regularized amortized optimization to the TSC XAI domain. Comprehensive empirical evaluations demonstrate that XMA is highly competitive with existing methods, attaining strong results in both explanation quality (faithfulness, robustness) and runtime efficiency across synthetic and real-world benchmarks.

## Impact Statement

Our research holds significant implications for critical domains relying on time series classification, such as finance and healthcare, where interpretability is a prerequisite for deployment. By achieving substantial improvements in computational efficiency, our work removes the historical resource barriers of XAI, enabling the seamless integration of explanations into real-time and resource-constrained applications.

From an ethical perspective, XMA empowers engineers to rapidly audit models, facilitating the timely detection of bias and fairness issues. In terms of societal consequences, this enhanced interpretability fosters greater public trust, allowing deep learning models to be adopted more safely in high-stakes environments.

However, we must also consider the potential risks associated with such efficiency. By drastically lowering the computational cost of explanation, our method could inadvertently accelerate the proliferation of complex autonomous systems at a pace that outstrips current regulatory frameworks or human oversight capabilities. The ability to easily "rationalize" model decisions might encourage the rapid deployment of powerful AI agents, potentially leading to scenarios where the scale and speed of automated decision-making become difficult to monitor or control effectively.

## Limitations

XMA shifts computational cost rather than eliminating it. Inference is cheap, but the one-time Stage-1 training (expert generation, per-instance selection, and gate training) takes about 37 minutes on MIT-ECG (Appendix D.2), dominated by expert generation and gate training. This cost is amortized over the test set, but it is not negligible for small deployments that explain few instances.

XMA's quality depends on the expert pool. The metric loss compensates only partially: when distillation is removed entirely, the metric loss alone reaches just $0.429$ AUPRC on MIT-ECG (Appendix C.11), so a pool of reasonable experts remains necessary as a warm start. Performance degrades gracefully when a single expert is removed (Appendix C.6), but we do not characterize behavior under a uniformly weak pool.

XMA optimizes faithfulness to the model, which is distinct from alignment with human semantics. On our benchmarks the two coincide: removing the features the model relies on also removes the features experts annotate as important (Appendix C.9). This need not hold for models that exploit spurious correlations, where a faithful explanation would instead reveal the mismatch rather than resolve it.

The faithfulness loss targets the true label $y_{\text{true}}$. On correctly classified instances this is immaterial, but on misclassified instances an explanation faithful to $y_{\text{true}}$ need not reflect the model's actual decision; we analyze this case in Appendix C.1 and recommend the predicted-label target when the goal is to diagnose model errors.

Our robustness evaluation covers Gaussian noise, FGSM perturbations, and missingness (Appendices C.8 and C.4). It does not cover time warping, which shifts feature positions and so requires a temporally-aware metric (e.g., DTW-based) rather than the position-sensitive Jaccard overlap used here; we leave this to future work.

The masking step size $\chi$ is fixed (default $0.1$). Results are insensitive to this choice (Appendix C.7), but an adaptive, instance-level step size could improve results further and is left to future work.

The robustness loss optimizes for stability under Gaussian perturbations and can in principle bias the explainer toward smoother, lower-variance attribution maps; the faithfulness component of the metric loss (Eq. 11) counters this pressure by penalizing maps that fail to identify decision-critical features, but the trade-off between stability and discriminativity is not fully characterized.

Finally, our evaluation uses cardiologist-verified annotations on MIT-ECG and proxy metrics on UEA, but not a controlled human-in-the-loop study. Broader expert-in-the-loop validation of whether the explanations aid practitioner decisions in real clinical workflows is an important next step (Liao & Varshney, 2021).

## Acknowledgements

We thank the anonymous reviewers and the area chair, whose feedback on terminology, ablation coverage, structural clarity, and presentation materially improved this paper.

We thank Anthony Bagnall, Eamonn Keogh, Jason Lines, Aaron Bostrom, James Large, Matthew Middlehurst, Amaia Abanda, Gustavo Batista, Mustafa Baydogan, Angus Dempster, Houtao Deng, Michael Flynn, Germain Forestier, Ben Fulcher, Tomasz Gorecki, Josef Grabocka, David Guijo-Rubio, Georgiana Ifrim, Hassan Ismail Fawaz, Myong Jeong, Rohit Kate, Jessica Lin, Benjamin Lucas, Pierre-Francois Marteau, Usue Mori, Abdullah Mueen, George Oastler, Alejandro Pasos Ruiz, Francois Petitjean, Thanawin Rakthanmanon, Patrick Schafer, Pavel Senin, Ahmed Shifaz, Diego Silva, Alexandra Stefan, Geoff Webb, Chang Wei Tan, Hoang Anh Dau, Kaveh Kamgar, Chin-Chia Michael Yeh, Yan Zhu, Shaghayegh Gharghabi, Chotirat Ann Ratanamahatana, and others of `timeseriesclassification.com` for producing and maintaining the UCR and UEA time series archives

used in this work.

We thank George Moody and Roger Mark for the MIT-BIH Arrhythmia Database, and Owen Queen, Thomas Hartvigsen, Teddy Koker, Huan He, Theodoros Tsiligkaridis, and Marinka Zitnik for the cardiologist-annotated MIT-ECG subset released with their NeurIPS 2023 work (Queen et al., 2023).

This research is part-funded by the European Union (Horizon Europe 2021–2027 Framework Programme, Grant Agreement number 10107245; views and opinions expressed are those of the author(s) only and do not necessarily reflect those of the European Union, which cannot be held responsible for them) and by the Engineering and Physical Sciences Research Council under grant number EP/X029174/1.

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

# A. Detailed Experimental Setup

## A.1. Datasets

**Synthetic Datasets.** We evaluate our framework using the synthetic dataset suite introduced by the SHAPEX authors (Huang et al., 2025), which draws methodological inspiration from (Germain et al., 2024). The suite comprises four distinct configurations: *Motif Count Classification Equal* (MCC-E), *Motif Type Classification Equal* (MTC-E), *Motif Count Classification High* (MCC-H), and *Motif Type Classification High* (MTC-H). The naming convention reflects the underlying task difficulty: (1) **Type vs. Count**: "Type" indicates that the class label is determined by the shape category of the inserted motif, whereas "Count" implies the label depends on the frequency of motif occurrences. (2) **Equal vs. High**: "Equal" signifies that the motif amplitude matches the baseline signal (making detection harder due to a low signal-to-noise ratio), while "High" indicates the motif amplitude significantly exceeds the background noise.

**MIT-ECG.** For real-world evaluation, we utilize the **MIT-ECG** dataset, a specialized subset of the MIT-BIH Arrhythmia Database introduced by Queen et al. in the TimeX framework (Queen et al., 2023). This dataset is particularly valuable for explainability research because, in addition to standard diagnostic labels for ECG instances, it includes dense, **cardiologist-verified ground truth annotations** that pinpoint the exact temporal regions responsible for the label. We refer readers to (Queen et al., 2023) for full details on the preprocessing and annotation protocols.

**UEA Datasets.** We conduct experiments on 11 multivariate time series datasets selected from the UEA Archive, as summarized in Table 2. These datasets were specifically chosen for their reliance on shape-based features, making them suitable benchmarks for evaluating saliency methods. Further details and data specifications can be found on the *timeseriesclassification.com* website. The selected datasets span HAR, Motion, and EEG types. Human Activity Recognition (HAR) datasets consist of multivariate time series data captured primarily through wearable inertial sensors (e.g., accelerometers, gyroscopes, magnetometers) embedded in smartphones, smartwatches, or specialized IMUs. Motion Trajectory (MOTION) datasets consist of multivariate time series recording the spatial coordinates $(x, y, z)$ or velocities of a specific entity over time, such as a hand, a pen tip, or an articulatory organ. Unlike HAR, which focuses on identifying activity states (e.g., walking vs. running), often using inertial measurements, MOTION tasks typically involve classifying the geometric shape or pattern of a movement trajectory. Finally, Electroencephalography (EEG) datasets consist of multivariate time series recording voltage fluctuations resulting from ionic current within the neurons of the brain.

For all datasets, we strictly adhere to the standard training and testing splits provided by the original authors. We train our objective-regularized amortized explainer network $g_\phi$ exclusively on the training subset and evaluate its performance on the hold-out test set. This protocol ensures that the model is never exposed to the test instances during the optimization phase, thereby preventing data leakage and guaranteeing a fair evaluation.

*Table 2.* Summary of Real-World Multivariate Time Series Datasets (UEA Archive) used in evaluation.

| Dataset Name | Length ($T$) | Variables ($d$) | Type |
|---|---|---|---|
| **UWaveGestureLibrary** | 315 | 3 | HAR |
| **ERing** | 65 | 4 | HAR |
| **RacketSports** | 30 | 6 | HAR |
| **NATOPS** | 51 | 24 | HAR |
| **CharacterTrajectories** | 182 | 3 | MOTION |
| **SelfRegulationSCP1** | 896 | 6 | EEG |
| **ArticularyWordRecognition** | 144 | 9 | MOTION |
| **Libras** | 45 | 2 | HAR |
| **BasicMotions** | 100 | 6 | HAR |
| **Cricket** | 1197 | 6 | HAR |
| **Epilepsy** | 206 | 3 | HAR |

## A.2. Hyper-parameters Choice

**Pool of "Experts".** To ensure robust supervision for our amortized framework, we curate diverse pools of expert explainers tailored to the specific characteristics of each dataset:

- **Synthetic Datasets:** We utilize the SHAPEX framework as the primary expert generator. To encourage diversity in the supervision signals, we vary the number of prototypes $K$ across the range $[1, 10]$, generating explanations with varying

degrees of shapelet complexity.

- **MIT-ECG:** We employ the MIX framework to generate a pool of 6 distinct experts. These are constructed by permuting the sliding window size over the set $\{24, 28, 36\}$ and the step size over $\{6, 9\}$, allowing the framework to capture anomalies at multiple temporal resolutions.

- **UEA Datasets:** We construct a heterogeneous expert pool comprising standard attribution methods: LIME, KernelSHAP, Integrated Gradients (IG), and Input×Gradient. Additionally, we introduce a variant termed *KeystoneIG*, which adapts the Keystone mechanism from MIX to calculate IG hierarchically (prioritizing the top ratio $R$ of features). We include KeystoneIG experts configured with five distinct ratios $R \in \{0.05, 0.10, 0.15, 0.20, 0.25\}$ to capture multi-scale importance.

**Ratios for Faithfulness Gate.** In theory, the faithfulness gate $h_\psi^R$ should be trained across the full spectrum of masking ratios $R \in [0, 1]$ to ensure global robustness. However, in practical interpretability evaluations, we are primarily interested in *sparse* explanations — specifically, the sufficiency of the Top-10% or Top-20% most salient features. Consequently, we configure the training to use high masking ratios $R \in \{0.8, 0.9\}$. By training the model to maintain prediction confidence even when 80% to 90% of the input is masked, we significantly reduce training costs while explicitly forcing the explainer $g_\phi$ to prioritize the identification of the most critical "top" features.

**Hyper-parameters for losses trade-off.** We provide the hyper-parameter options in Tab 3.

*Table 3.* Hyper-parameter settings for the loss function components across different datasets. $\lambda$ controls the balance between expert distillation and metric optimization, while $\beta$ balances the trade-off between faithfulness and robustness constraints.

| Dataset | $\lambda$ (Distillation vs. Metrics) | $\beta$ (Faithfulness vs. Robustness) |
|---|---|---|
| **Synthetic** | 1.0 | 0.5 |
| **MIT-ECG** | 0.001 | 100.0 |
| **UEA** | 1.0 | 1.0 |

### A.3. Objective-Regularized Amortized Deep Neural Networks

**DNN Architectures.** To demonstrate the efficiency of our framework, we deliberately employ lightweight architectures for the explainer network $g_\phi$. For the synthetic datasets, we utilize a compact CNN architecture augmented with residual skip connections. For the real-world benchmarks (MIT-ECG and UEA), we employ a standard Fully Convolutional Network (FCN). Notably, despite the simplicity of these architectures, our method achieves state-of-the-art explainability performance in Time Series Classification (TSC). This finding highlights the effectiveness of the objective-regularized amortized paradigm, proving that high-fidelity explanations can be generated with minimal computational overhead without relying on over-parameterized models.

**Training Hyper-parameters.** We optimize all models using the Adam optimizer. The specific configurations for each dataset are as follows:

- **Synthetic Datasets:** Learning rate $\alpha = 5 \times 10^{-4}$, batch size $B = 32$, and weight decay $\lambda_{wd} = 10^{-5}$.

- **MIT-ECG:** Learning rate $\alpha = 1 \times 10^{-4}$ and batch size $B = 16$.

- **UEA Datasets:** Learning rate $\alpha = 1 \times 10^{-2}$ and batch size $B = 16$.

### A.4. Hardware Environment

All experiments were conducted on a high-performance workstation running Ubuntu 24.04.3 LTS, equipped with 64GB of system memory (RAM) and dual NVIDIA RTX A5000 GPUs.

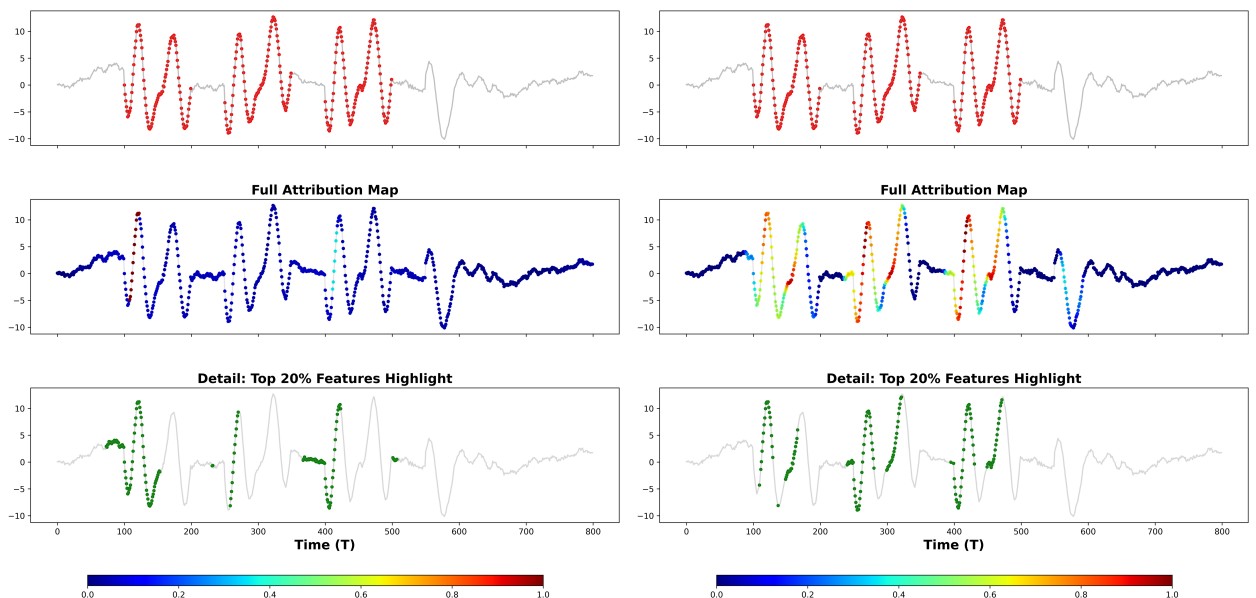

*Figure 5.* **Qualitative visualization on Synthetic Data MCC-E.** We compare the baseline ShapeX (Left) against our proposed XMA framework (Right). **(Top)** The raw time series with human-annotated ground truth motifs highlighted in **red**. **(Middle)** The full attribution maps generated by each method, where color indicates importance. **(Bottom)** The top 20% most significant features identified by each explainer (green). Observe that **XMA (Right)** successfully recovers the continuous motif structure, aligning closely with the **red ground truth** in the top row. In contrast, ShapeX (Left) produces fragmented explanations that fail to capture the complete pattern, demonstrating XMA's superior ability to preserve semantic faithfulness.

## B. Experimental Results

### B.1. Qualitative Results for Synthetic Datasets

We present a comprehensive visual comparison between our proposed XMA framework and the state-of-the-art baseline, ShapeX, across all four synthetic configurations. The qualitative results are illustrated in Figures 5, 6, 7, and 8.

As visually demonstrated, our method consistently produces attribution maps that are more semantically faithful to the ground truth than the baseline. Specifically, while ShapeX often yields fragmented explanations that struggle to delineate the full boundaries of the patterns, XMA successfully recovers the continuous structure of the motifs. This is evident in the precise alignment between our identified top features (Green) and the human-annotated ground truth (Red) across all dataset variations (MTC/MCC, High/Equal), confirming XMA's superior localization capabilities even in challenging low signal-to-noise environments.

### B.2. Qualitative Results for MIT-ECG Datasets

We visualize the attribution results on the MIT-ECG dataset in Figure 9. The results demonstrate that our proposed **Faithfulness-Preserving Segmentation** mechanism effectively translates continuous attribution scores into discrete, interpretable segments. Crucially, the high-importance segments identified by our method (highlighted in red) consistently align with the cardiologist-verified ground truth anomalies. This confirms that treating segments as the fundamental unit of explanation not only simplifies the visual output but also ensures that the explanations remain clinically meaningful and human-aligned.

### B.3. Comparison with Recent TSC Explainers

We compare XMA against two recent subsequence-based explainers, TIMING (Jang et al., 2025) and Implet (Meng et al., 2025), on MIT-ECG (Table 4) and Libras (Table 5).

On MIT-ECG, XMA attains the highest AUPRC (0.890, versus 0.552 for Implet and 0.468 for TIMING). TIMING reaches the highest AUP (0.817) but an AUR of only 0.096, recovering under 10% of the cardiologist-annotated regions; such low

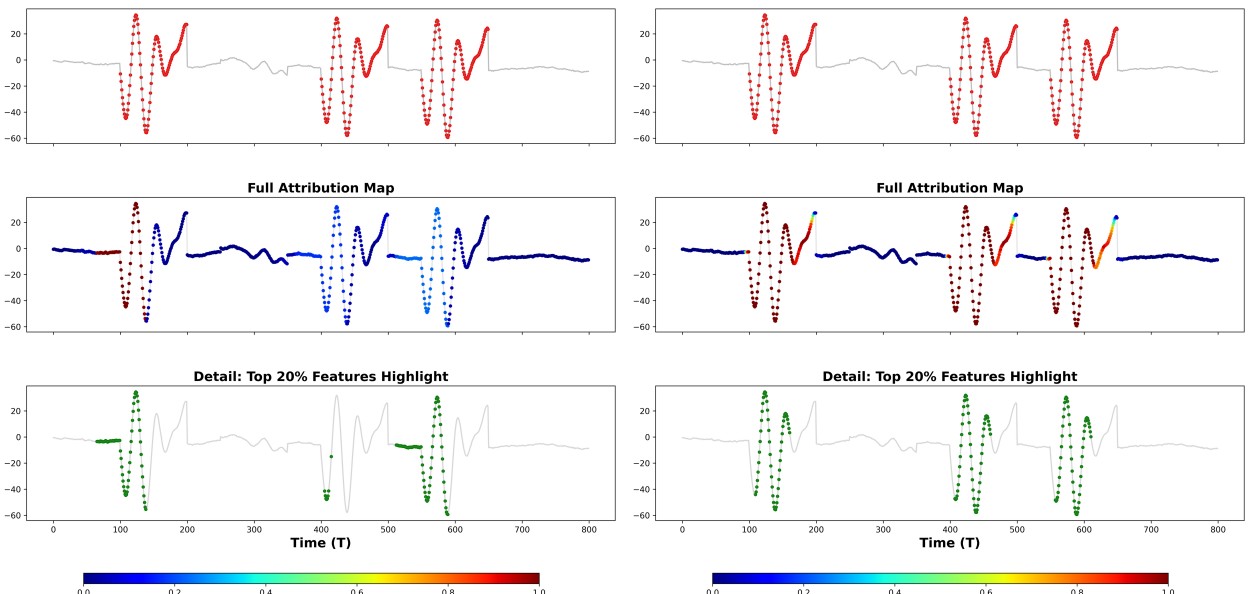

*Figure 6.* **Qualitative visualization on Synthetic Data MCC-H.** We compare the baseline ShapeX (Left) against our proposed XMA framework (Right). **(Top)** The raw time series with human-annotated ground truth motifs highlighted in **red**. **(Middle)** The full attribution maps generated by each method, where color indicates importance. **(Bottom)** The top 20% most significant features identified by each explainer (green). Observe that **XMA (Right)** successfully recovers the continuous motif structure, aligning closely with the **red ground truth** in the top row. In contrast, ShapeX (Left) produces fragmented explanations that fail to capture the complete pattern, demonstrating XMA's superior ability to preserve semantic faithfulness.

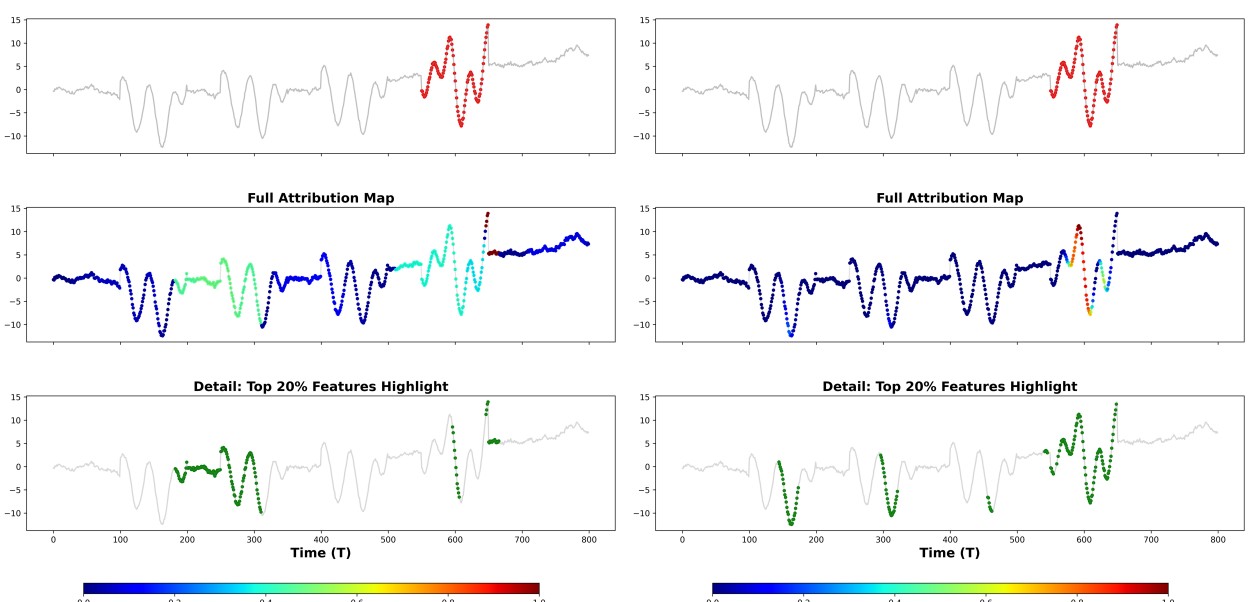

*Figure 7.* **Qualitative visualization on Synthetic Data (MTC-E).** We compare the baseline ShapeX (Left) against our proposed XMA framework (Right) on the Motif Type Classification-Equal dataset. **(Top)** The raw time series with human-annotated ground truth motifs highlighted in **red**. **(Middle)** The full attribution maps generated by each method, where color intensity indicates feature importance (red/warm colors denote higher importance). **(Bottom)** The discrete selection of the Top 20% most significant features (highlighted in **green**). Observe that **XMA (Right)** successfully recovers the continuous motif structure, with the green selection aligning closely with the **red ground truth** segments in the top row. In contrast, ShapeX (Left) produces fragmented explanations that fail to capture the complete pattern, highlighting XMA's superior ability to preserve semantic faithfulness in low signal-to-noise environments.

recall would be insufficient for clinical decision support. XMA maintains a balanced precision–recall profile (AUP 0.594, AUR 0.877).

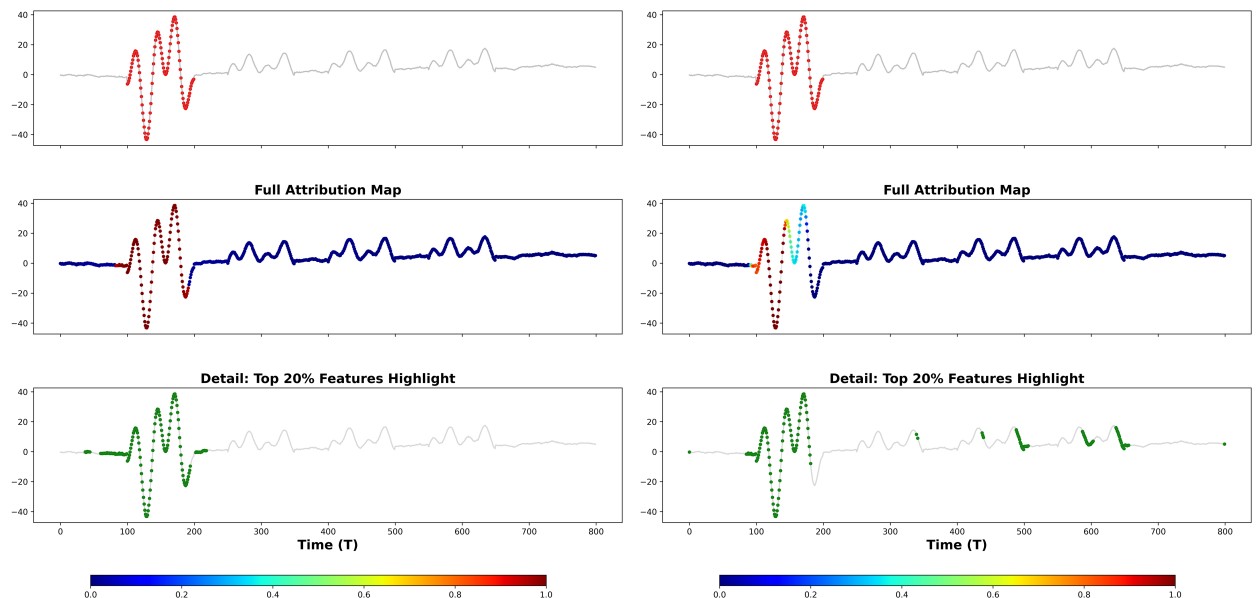

*Figure 8.* **Qualitative visualization on Synthetic Data MTC-H.** We compare the baseline ShapeX (Left) against our proposed XMA framework (Right). **(Top)** The raw time series with human-annotated ground truth motifs highlighted in **red**. **(Middle)** The full attribution maps generated by each method, where color indicates importance. **(Bottom)** The top 20% most significant features identified by each explainer (green). Observe that **XMA (Right)** successfully recovers the continuous motif structure, aligning closely with the **red ground truth** in the top row. In contrast, ShapeX (Left) produces a good explanation in this case because the high-signal motifs are easily recognized.

*Table 4.* Comparison on MIT-ECG. XMA attains the best AUPRC and a balanced precision–recall profile. Best per metric in bold.

| Method | AUPRC | AUP | AUR |
|--------|-------|-----|-----|
| XMA | **0.890** | 0.594 | **0.877** |
| Implet | 0.552 | 0.429 | 0.853 |
| TIMING | 0.468 | **0.817** | 0.096 |

On Libras, XMA attains the highest faithfulness across all three backbones. Implet is more robust on PatchTST (0.759 vs. 0.622), while XMA is most robust on FCN and ResNet.

## C. Experimental Analysis

### C.1. True versus Predicted Label for Faithfulness

The faithfulness loss (Section 3.2) is defined with respect to the true label $y_{\text{true}}$. We default to $y_{\text{true}}$ because the standard AUC-based faithfulness protocol implicitly assumes $y_{\text{true}} = y_{\text{pred}}$, which holds for the large majority of correctly predicted instances; the choice matters only on the misclassified minority. In both IMEKD selection and ORAOE training, misclassified instances are scored and trained against $y_{\text{true}}$ like any other instance — we do not give them special treatment. We analyze here how this choice affects evaluation, separating correctly predicted instances (where $y_{\text{true}} = y_{\text{pred}}$ and the choice is immaterial) from misclassified instances (where the two targets diverge).

**Overall effect.** On MIT-ECG, switching the faithfulness target between $y_{\text{true}}$ and $y_{\text{pred}}$ changes the aggregate metrics only marginally (Table 6), by less than 5% on all three metrics. Because $y_{\text{true}} = y_{\text{pred}}$ on correctly classified instances, the distinction has no effect for the large majority of the evaluation set.

**Analysis on misclassified instances.** The distinction matters only where the model errs. Let $T$ and $P$ denote the top-20% salient features from the true-label and predicted-label explanations, respectively. For each feature subset, we measure the *flip rate*: the fraction of misclassified instances whose prediction changes when only that subset is retained (Table 7).

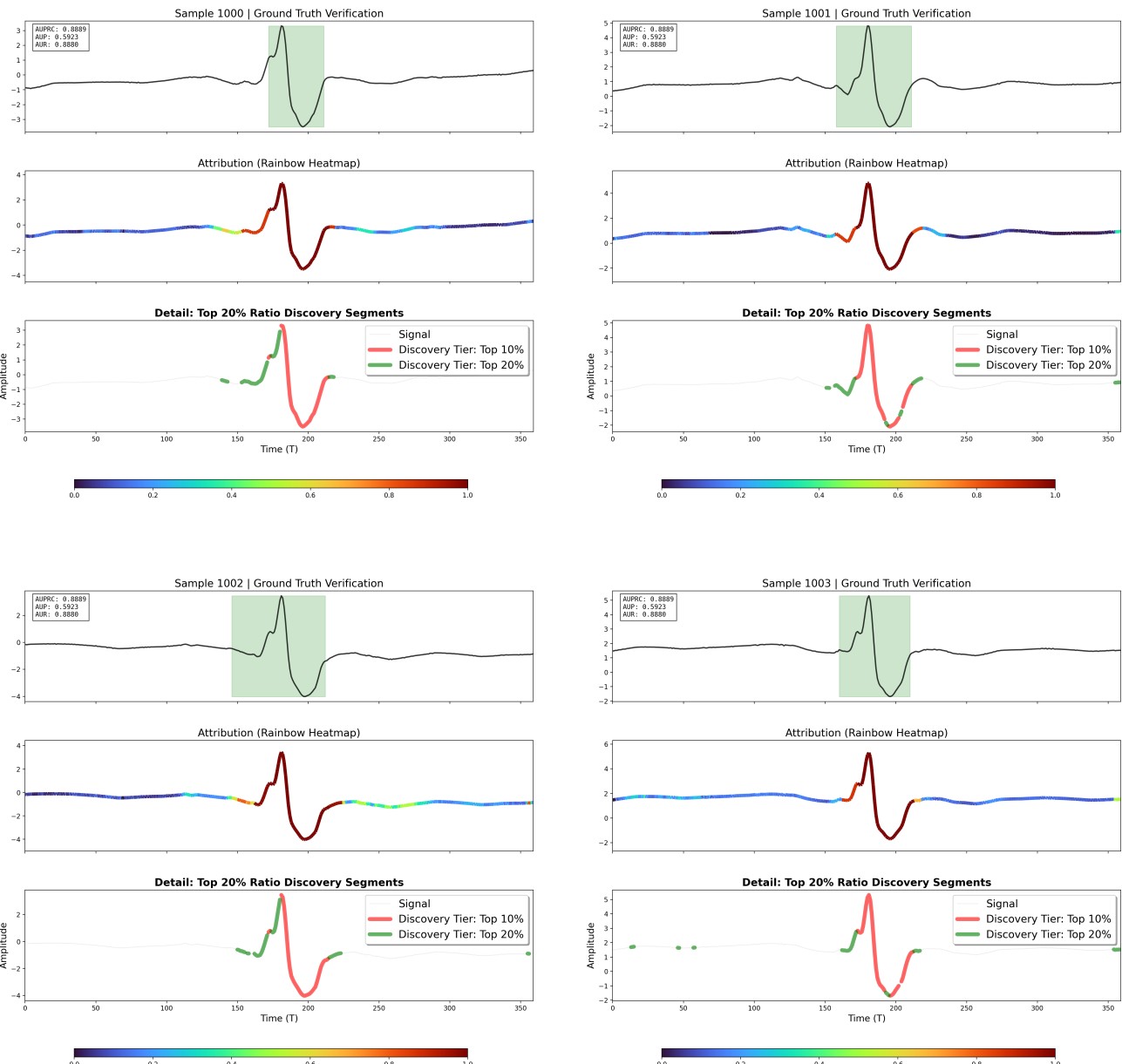

*Figure 9.* **Faithfulness-Preserving Segmentation across diverse samples on MIT-ECG.** We visualize the segmentation results on four distinct heartbeats from the MIT-ECG dataset to demonstrate the method's consistency. For each sample: **(Top)** The ground truth verified by cardiologists, with the critical anomaly highlighted in the **green box**. **(Middle)** The continuous attribution map (rainbow heatmap), where red indicates high importance. **(Bottom)** Our **Faithfulness-Preserving Segmentation** for step 10% result. Across all four diverse signal patterns, our method consistently identifies the expert-verified anomaly, assigning it to the highest importance tier (**Red: Top 10%**). This confirms that our segmentation algorithm reliably translates complex attribution maps into precise, interpretable segments regardless of signal variation.

Three observations follow. First, the two perspectives overlap substantially: the shared set $T \cap P$ alone yields a $38.1\%$ flip rate, so the explanations largely agree even on misclassified instances. Second, features unique to the predicted-label explanation ($P \setminus T$) have the least influence ($9.1\%$), suggesting that the model's errors arise from subtle weighting differences rather than attending to entirely wrong features. Third, features unique to the true-label explanation ($T \setminus P$) carry nearly twice that influence ($17.1\%$), representing semantically relevant regions that the model underweighted in its incorrect decision.

**Practical implication.** The two targets serve complementary purposes. True-label explanations help practitioners audit

*Table 5.* Faithfulness and robustness on Libras across three backbones. XMA attains the highest faithfulness on all three; Implet is more robust on PatchTST. Best per column group in bold.

| Backbone | Faithfulness | | | Robustness | | |
|---|---|---|---|---|---|---|
| | XMA | TIMING | Implet | XMA | TIMING | Implet |
| FCN | **0.937** | 0.925 | 0.922 | **0.826** | 0.723 | 0.776 |
| PatchTST | **0.974** | 0.887 | 0.886 | 0.622 | 0.482 | **0.759** |
| ResNet | **0.946** | 0.896 | 0.878 | **0.714** | 0.588 | 0.595 |

*Table 6.* XMA faithfulness metrics on MIT-ECG under true-label versus predicted-label targets. Differences are below 5% on all metrics.

| Metric | True label | Predicted label | Difference |
|---|---|---|---|
| AUPRC | 0.890 | 0.884 | $-0.006$ |
| AUP | 0.594 | 0.570 | $-0.024$ |
| AUR | 0.877 | 0.892 | $+0.015$ |

whether the model attends to semantically meaningful features, while predicted-label explanations help developers diagnose failure cases. A complete account of model behavior on misclassified instances benefits from both. This complementary use of true- and predicted-label targets follows the functional-information perspective on model interpretation (Gat et al., 2022).

## C.2. Expert Selection Diversity and Metric Sensitivity

This section provides evidence for the claim that a single best expert varies across instances. We emphasize that this is an empirical observation under our composite faithfulness-and-robustness criterion (Section 3.1), not a theoretical guarantee that any finite pool contains the optimal attribution.

**Selection diversity.** Tables 8 and 9 report how often each expert is chosen as the instance-best on MIT-ECG and Libras (FCN). No expert dominates: on MIT-ECG selection rates span $14.94\%$–$30.62\%$ across five MIX configurations, and on Libras they span $1.7\%$–$49.4\%$ across the five experts. Different instances benefit from different experts, which is the premise of instance-level selection.

The two distributions differ substantially: MIT-ECG selects fairly evenly across configurations, whereas Libras concentrates on LIME. This shows that the useful expert set is dataset-dependent, so committing to one global explainer would be suboptimal.

**Metric sensitivity.** The selected expert also depends on the scoring criterion. Table 10 reports how often two criteria agree on the same expert on Libras. Faithfulness-only and robustness-only criteria agree for only about $11\%$ of instances. We read this not as instability but as evidence that faithfulness and robustness measure different aspects of explanation quality: the most faithful expert for an instance is rarely the most robust. This motivates the composite criterion, which balances both.

Despite this selection-level sensitivity, the final explanation quality is stable: the masking-step sensitivity analysis (Appendix C.7) shows under $1\%$ variation, and the remove-one-expert ablation (Appendix C.6) shows under $3\%$ degradation. The amortized training with the metric loss absorbs selection-level variation into stable final explanations.

## C.3. Sensitivity to the Masking Baseline

The faithfulness loss (Eq. 6) and the evaluation metric (Eq. 12) both replace masked features with a baseline signal $\mathbf{x}'$. By default we set $\mathbf{x}'$ to the per-feature mean of the training set, which is approximately zero under per-dataset z-score normalization. The baseline is dataset-specific by construction: each dataset uses its own training-mean baseline, held fixed across all compared methods on that dataset. To test whether our results depend on this choice, we keep the trained explainer fixed and vary only the evaluation baseline on MIT-ECG (Table 11).

The metrics are stable across all three baselines: AUPRC ranges over 0.889–0.893 and AUR over 0.876–0.877, so the reported faithfulness is not an artifact of the specific baseline signal. AUP shows the largest variation (0.594–0.619), indicating mild sensitivity of the precision component to the baseline, though it does not change the ranking of XMA against

*Table 7.* Flip rate on misclassified MIT-ECG instances for feature subsets derived from the true-label ($T$) and predicted-label ($P$) top-20% features obtained by explanations.

| Retained subset | Flip rate | Interpretation |
|---|---|---|
| $T \cap P$ | 38.1% | shared, decision-critical |
| $T$ | 45.7% | all true-label features |
| $P$ | 45.7% | all predicted-label features |
| $T \setminus P$ | 17.1% | true-only |
| $P \setminus T$ | 9.1% | predicted-only |

*Table 8.* Expert selection rate on MIT-ECG, across five MIX configurations (window size $ws$, step size $s$).

| Expert | Selected (%) |
|---|---|
| MIX ($ws$=24, $s$=9) | 30.62 |
| MIX ($ws$=24, $s$=6) | 20.84 |
| MIX ($ws$=36, $s$=9) | 17.77 |
| MIX ($ws$=36, $s$=6) | 15.84 |
| MIX ($ws$=28, $s$=6) | 14.94 |

the compared methods.

### C.4. Robustness under Perturbations

Beyond Gaussian noise, we evaluate robustness under missingness, which is genuinely distinct from Gaussian corruption and directly relevant to deployment, where sensor dropout and incomplete recordings are common. We do not evaluate jitter, which is equivalent to Gaussian noise, or time warping, which shifts feature positions and so requires a temporally-aware metric (e.g., DTW-based) rather than the Jaccard overlap used here; we leave the latter for future work.

**Gaussian-trained model under missingness.** Table 12 evaluates the Gaussian-trained explainer under $20\%$ missingness (randomly dropping time steps and replacing them with the baseline) on Libras. Robustness drops substantially relative to Gaussian noise across all backbones, confirming that missingness is the stronger perturbation: removing the signal entirely is more disruptive than corrupting it.

**Training with missingness.** We also train an explainer with missingness perturbations and compare it against the Gaussian-trained explainer under both perturbation types (Table 13). Missingness training improves robustness to missingness but reduces robustness to Gaussian noise and slightly lowers faithfulness, whereas Gaussian training gives more uniform robustness across both perturbation types. This comparison is straightforward to run because the amortized explainer is retrained once, unlike per-instance explainers that would require re-optimization for every test point.

### C.5. Seed Stability

We assess stability across three random seeds on Libras (Table 14). Faithfulness standard deviation stays below $0.7\%$ and robustness below $1.5\%$ across all backbones, indicating that the reported results are not seed-dependent. Unless otherwise stated, ROC-AUC for multi-class datasets is computed as macro-averaged one-vs-rest.

### C.6. Pool Size and Expert Removal

We examine how pool composition affects explanation quality on Libras: how quality scales with pool size $K$, and how it degrades when a selected expert is removed.

**Effect of pool size.** Table 15 varies $K$ on Libras (FCN). The largest gain comes from $K$=1 to $K$=2: faithfulness rises from 0.926 to 0.939 (+1.3%) and robustness from 0.695 to 0.820 (+12.5%). Adding the third expert improves faithfulness slightly further (0.942) while holding robustness, indicating diminishing returns beyond a small pool. A small, diverse pool (roughly $K$=3 to 5) therefore captures most of the benefit without the cost of running many experts in Stage 1.

**Effect of removing a selected expert.** We remove one expert during training, then evaluate only on the instances where

*Table 9.* Expert selection rate on Libras (FCN).

| Expert | Selected (%) |
|---|---|
| LIME | 49.4 |
| IG variants | 35.6 |
| IG | 9.4 |
| Input×Gradient | 3.9 |
| KernelSHAP | 1.7 |

*Table 10.* Agreement rate (% of instances selecting the same expert) between scoring criteria on Libras.

| Criteria compared | FCN | PatchTST | ResNet |
|---|---|---|---|
| Faithfulness vs. Composite | 30.6 | 27.8 | 29.4 |
| Robustness vs. Composite | 65.0 | 55.6 | 56.7 |
| Faithfulness vs. Robustness | 12.8 | 11.1 | 10.6 |

that expert would have been selected, the most adversarial setting for the reduced pool. Table 16 reports faithfulness and robustness for removing IG and Input×Gradient. On faithfulness, the full pool wins in 4 of 6 comparisons; on robustness the two pools are close, with the full pool ahead in 3 of 6. The degradation is modest throughout (within about 1–3%), so the framework degrades gracefully: the remaining experts and the metric loss still supply enough signal to learn reasonable attributions without the best teacher.

The full pool's advantage is clearest on faithfulness, the quantity expert selection directly optimizes. Robustness is closer between the two pools and occasionally favors the reduced one, which is consistent with the metric loss, not the teacher set, being the main driver of attribution stability. In the few cases where the reduced pool scores marginally higher (within 1%), removing a weaker teacher leaves the metric loss more room to refine beyond imitation. This is consistent with the reverse ablation (Appendix C.11), where metric-only training without any teacher produces poor explanations (0.429 AUPRC on MIT-ECG). The expert pool prevents blind optimization, while the metric loss prevents the student from being bounded by pool quality.

**Removing perturbation-based experts.** We repeat the analysis for the perturbation-based experts, LIME and KernelSHAP (Table 17), again evaluating only on instances where the removed expert would have been selected. The full pool wins on faithfulness in 4 of 6 comparisons. Two cases favor the reduced pool: PatchTST without LIME (0.673 vs. 0.668) and FCN without KernelSHAP (0.485 vs. 0.445). The KernelSHAP case is the largest such gap, but it arises on a very small subset - KernelSHAP is selected for only 1.7% of Libras instances - where the metric loss can refine a weak warm start more easily. We therefore do not claim that more experts always help; rather, a diverse pool helps on average, and the metric loss compensates when an individual teacher is weak or rarely selected.

### C.7. Sensitivity to the Masking Step Size

The faithfulness loss uses a masking step size $\chi$ (default 0.1; Section 3.2). We test sensitivity to this choice in two ways on Libras: varying $\chi$ at training time (Table 18), and varying the evaluation step $\delta$ while holding training fixed at $\chi = 0.1$ (Table 19). Separating the two distinguishes whether the model overfits to the training granularity from whether the evaluation itself is granularity-sensitive.

As shown in Tables 18 and 19, faithfulness varies by at most about 1% across all tested step sizes for every architecture, and no single value dominates: FCN slightly favors $\chi = 0.05$ while PatchTST slightly favors $\chi = 0.2$. The training-time and evaluation-time results are nearly identical, differing only marginally for PatchTST (e.g., 0.954 vs. 0.955 at the default step), which indicates that the explainer does not overfit to the $\chi = 0.1$ training granularity. An adaptive, instance-level step size could improve results further; we note this as future work.

### C.8. Adversarial Robustness under FGSM

We test whether the explainer's robustness extends from Gaussian noise to adversarial perturbations. We train two XMA variants on Libras, one with the Gaussian robustness loss (Original) and one replacing it with FGSM adversarial perturbations

*Table 11.* Faithfulness metrics on MIT-ECG under three masking baselines, with the trained explainer held fixed. AUPRC varies by less than $0.4\%$ and AUR by less than $0.1\%$ across all choices.

| Baseline | AUPRC | AUP | AUR |
|---|---|---|---|
| Mean (default) | 0.890 | 0.594 | 0.877 |
| Gaussian noise | 0.893 | 0.619 | 0.876 |
| Low-pass filtered | 0.889 | 0.598 | 0.876 |

*Table 12.* Robustness of the Gaussian-trained explainer under Gaussian noise versus $20\%$ missingness, on Libras.

| Architecture | Rob. (Gaussian) | Rob. (Missingness) |
|---|---|---|
| FCN | 0.823 | 0.543 |
| PatchTST | 0.697 | 0.561 |
| ResNet | 0.707 | 0.551 |

(FGSM), and evaluate both under FGSM attacks and random noise (Table 20). Because XMA's explainer is a differentiable network, adversarial perturbations integrate directly into its training loop, which is impractical for per-instance explainers such as LIME or KernelSHAP.

Three findings emerge. First, the Gaussian-trained explainer stays reasonably stable under FGSM attacks without any adversarial training, and on PatchTST it is even more FGSM-robust than the FGSM-trained variant (0.715 vs. 0.599). Second, FGSM training improves adversarial robustness on FCN and ResNet but not on PatchTST, and it tends to lower faithfulness (e.g., ResNet drops from 0.946 to 0.926). Third, this points to a non-trivial trade-off between adversarial robustness and faithfulness that warrants dedicated study. We treat this as a preliminary exploration enabled by the amortized design; stronger attacks (e.g., PGD) and time-series-specific perturbations are left to future work.

### C.9. Faithfulness versus Human Interpretability

XMA optimizes faithfulness to the model's decision process, which is distinct from alignment with human semantic understanding: a model may rely on features that are predictive but not meaningful to a human. We test whether these two notions coincide on our benchmarks by removing the features that ground truth marks as important (cardiologist annotations for MIT-ECG, known motifs for MCCE) and measuring the effect on the model. If the model relies on the same features that experts consider meaningful, removing them should degrade performance toward random.

As shown in Tables 21 and 22, removing ground-truth important features collapses model performance to the random level on both datasets: accuracy falls from 0.854 to 0.440 on MIT-ECG (with confidence dropping from 0.840 to 0.441) and from 0.985 to 0.499 on MCCE. Removing an equal portion of redundant features leaves performance almost unchanged (0.843 accuracy on MIT-ECG, 0.894 on MCCE). In these domains, the model's decision-relevant features therefore coincide with the features experts identify as meaningful, so faithfulness to the model also reflects human-relevant structure. This alignment is not guaranteed in general: a model trained on data with spurious correlations could rely on features that are predictive but not semantically meaningful, in which case a faithful explanation would expose that mismatch rather than resolve it.

### C.10. Sensitivity to the Robustness Threshold

Robustness is measured by the Jaccard overlap of the top salient features, binarized at a threshold (default $20\%$; Section 4.1). We follow the $20\%$ convention used in recent TSC XAI evaluation (Jang et al., 2025; Liu et al., 2024b). To verify that this choice does not bias the comparison, Table 23 reports robustness across thresholds from $10\%$ to $30\%$ on Libras (FCN).

The method ranking is stable across all thresholds: XMA and IG occupy the top two positions, while LIME and KernelSHAP remain far lower. XMA and IG are within 0.01 at $10\%$ (IG marginally ahead), and XMA leads from $15\%$ onward with a widening margin at higher thresholds. No threshold reverses the overall ranking, so the main-result comparison is not an artifact of the $20\%$ choice.

*Table 13.* Faithfulness and robustness of explainers trained with missingness versus Gaussian perturbations, on Libras. Best per architecture in bold.

| Architecture | Training | Faith. | Rob. (Miss.) | Rob. (Gauss.) |
|---|---|---|---|---|
| FCN | Missingness | 0.933 | **0.666** | 0.814 |
| | Gaussian | **0.938** | 0.555 | **0.820** |
| PatchTST | Missingness | **0.962** | 0.471 | 0.559 |
| | Gaussian | 0.962 | **0.577** | **0.719** |
| ResNet | Missingness | 0.920 | **0.627** | 0.717 |
| | Gaussian | **0.946** | 0.537 | **0.682** |

*Table 14.* Faithfulness and robustness across three random seeds on Libras (mean $\pm$ standard deviation).

| Architecture | Faithfulness | Robustness |
|---|---|---|
| FCN | $0.941 \pm 0.001$ | $0.822 \pm 0.007$ |
| PatchTST | $0.970 \pm 0.004$ | $0.624 \pm 0.013$ |
| ResNet | $0.939 \pm 0.007$ | $0.714 \pm 0.015$ |

### C.11. Ablation: Necessity of Distillation

Section 4.4 reports the reverse ablation on AUPRC. Here we give the full per-metric results on MIT-ECG (Table 24) and the per-backbone results on Libras (Table 25), comparing the full model, the distillation-only variant (XM), and the metric-only variant (no distillation).

Without expert distillation, the explainer must discover attributions from the faithfulness and robustness gradients alone, a weak signal with many poor local minima: AUPRC falls to 0.429 on MIT-ECG. Distillation supplies the warm start that makes the metric loss effective. The metric loss in turn lets the explainer refine beyond the teacher: on Libras, the full model improves faithfulness over the metric-only variant on all three backbones. The one exception is PatchTST robustness, where the metric-only variant is higher (0.690 vs. 0.622), indicating that distillation can occasionally constrain robustness optimization; the faithfulness gains outweigh this on balance. The two components are therefore complementary: distillation provides a high-quality warm start, and the metric loss refines the explanation beyond teacher quality.

## D. Algorithm Analysis

### D.1. Computational Complexity Analysis

Let $C(\mathbf{x}, f_\theta)$ denote the computational cost of a forward pass through the target DNN $f_\theta$ for an input $\mathbf{x}$, and $C_p(\mathbf{x}, f_\theta)$ denote the cost of a backpropagation pass. We assume the standard approximation $C(\mathbf{x}, f_\theta) \simeq C_p(\mathbf{x}, f_\theta)$. Additionally, let $C_{XMA}(\mathbf{x}, g_\phi)$ represent the inference complexity of our proposed explainer network with the input-only $\mathbf{x}$, and $C_{1D-CNN}(\mathbf{x}, f_{SHAPEX})$ represent the complexity of the 1D-CNN used in SHAPEX.

**Time Complexity for SHAPEX.** SHAPEX operates in two distinct stages: (1) Shapelet generation, and (2) Attribution scoring via SHAP. First, a forward pass through the auxiliary 1D-CNN generator, denoted as $C_{1D-CNN}(\mathbf{x}, f_{SHAPEX})$, is required to obtain candidate shapelets. Second, to compute the contribution of each shapelet, the method employs a SHAP-based estimation which necessitates querying the target DNN. Let $N_s$ denote the number of extracted subsequences (or shapelets) that require scoring. Since SHAP estimation relies on evaluating the target model's output for varying subsets of features, the complexity is proportional to the number of subsequences multiplied by the target model's inference cost. Thus, the total complexity is:

$$\mathfrak{C}_{SHAPEX} \approx C_{1D-CNN}(\mathbf{x}, f_{SHAPEX}) + N_s \cdot C(\mathbf{x}, f_\theta) \tag{13}$$

This linear dependence on $N_s$ (which can be large for long time series) creates a significant computational bottleneck compared to direct inference methods.

**Time Complexity for MIX.** The MIX framework employs Integrated Gradients (IG) as its core attribution mechanism. Approximating the path integral for IG requires $n$ discrete steps, necessitating $n$ queries to the model $f_\theta$. To capture

*Table 15.* Effect of pool size $K$ on Libras (FCN). The largest gain is from $K=1$ to $K=2$; the third expert adds little.

| Pool size | Faithfulness | Robustness |
|---|---|---|
| $K = 1$ (single best expert) | 0.926 | 0.695 |
| $K = 2$ | 0.939 | **0.820** |
| $K = 3$ (full pool) | **0.942** | 0.817 |

*Table 16.* Remove-one-expert results on Libras, evaluated only on instances where the removed expert would have been selected. Best per architecture in bold.

| Removed | Architecture | Faithfulness | | Robustness | |
|---|---|---|---|---|---|
| | | Full | Reduced | Full | Reduced |
| | FCN | **0.673** | 0.662 | 0.835 | **0.844** |
| IG | PatchTST | **0.804** | 0.797 | **0.635** | 0.628 |
| | ResNet | 0.713 | **0.718** | 0.688 | 0.688 |
| | FCN | **0.596** | 0.586 | 0.771 | **0.779** |
| Input×Gradient | PatchTST | **0.803** | 0.798 | **0.616** | 0.584 |
| | ResNet | 0.685 | **0.691** | **0.701** | 0.687 |

robust features, this process is repeated across $N_V$ distinct augmented views, scaling the initial cost to $N_V \cdot n \cdot C_p(\mathbf{x}, f_\theta)$. Subsequently, MIX performs an iterative interaction phase to evaluate faithfulness metrics and select the optimal explanation. Let $N_I$ denote the total number of inference queries required for this metric-based evaluation. The overall time complexity is therefore dominated by the sum of these repeated model interactions:

$$\mathfrak{C}_{MIX} \approx (N_V \cdot n + N_I) \cdot C_p(\mathbf{x}, f_\theta) \tag{14}$$

The multiplicative factor $N_V \cdot n$ renders MIX computationally expensive, particularly for applications requiring high-resolution path integrals or multiple robust views.

**Time Complexity for XMA (Ours).** The XMA framework is designed for efficiency, avoiding iterative optimization loops during inference. It first requires a single backpropagation pass through the target model to extract gradient information $\mathbf{I}_{grad}$ for feature enrichment, incurring a cost of $C_p(\mathbf{x}, f_\theta)$. Subsequently, the explainer network $g_\phi$ processes the enriched input in a purely amortized manner. The total computational complexity is thus the sum of one gradient extraction and one forward pass of the explainer:

$$\mathfrak{C}_{XMA} \approx C_{XMA}(\text{Concat}(\mathbf{x}, \mathbf{I}_{grad}), g_\phi) + C_p(\mathbf{x}, f_\theta) \tag{15}$$

Unlike MIX or SHAPEX, XMA does not depend on the number of evaluation steps or shapelet candidates, making it significantly faster for real-time deployment.

**Time Complexity for Input-Only XMA.** The Input-Only XMA variant removes the gradient input, thereby decoupling the explanation process from the target model's computational graph. Its inference time is independent of the target DNN $f_\theta$ and relies solely on the forward pass of the explainer network $g_\phi$ processing the raw input $\mathbf{x}$:

$$\mathfrak{C}'_{XMA} \approx C_{XMA}(\mathbf{x}, g_\phi) \tag{16}$$

By eliminating the need for backpropagation $C_p(\mathbf{x}, f_\theta)$, Input-Only XMA achieves XMA's theoretical lower bound on latency, making it highly suitable for resource-constrained real-time applications.

**D.2. End-to-End Computational Cost**

XMA shifts cost from inference to a one-time training phase. Table 26 reports the full pipeline cost on MIT-ECG (NVIDIA A5000), separating the one-time Stage-1 and Stage-2 training from per-test-set inference; inference timings for MIX, LIME, and TIMING are included for comparison.

*Table 17.* Remove-one-expert faithfulness on Libras for the perturbation-based experts, evaluated only on instances where the removed expert would have been selected. Best per architecture in bold.

| Removed | Architecture | Full pool | Reduced |
|---|---|---|---|
| LIME | FCN | **0.706** | 0.678 |
| | PatchTST | 0.668 | **0.673** |
| | ResNet | **0.548** | 0.529 |
| KernelSHAP | FCN | 0.445 | **0.485** |
| | PatchTST | **0.745** | 0.709 |
| | ResNet | **0.750** | 0.743 |

*Table 18.* Faithfulness on Libras with varying training step size $\chi$. Variation is about $1\%$ or less per architecture.

| Architecture | $\chi = 0.05$ | $\chi = 0.1$ | $\chi = 0.2$ |
|---|---|---|---|
| FCN | 0.936 | 0.929 | 0.927 |
| PatchTST | 0.961 | 0.954 | 0.965 |
| ResNet | 0.923 | 0.934 | 0.932 |

Stage 1 (expert generation, per-instance selection, and gate training) and Stage 2 (explainer training) together take about 37 minutes. This is on the same order of magnitude as training the TSC classifier itself under the standard protocol, so the one-time XMA overhead is comparable to fitting the model being explained. This cost is incurred once: the trained explainer then attributes the entire MIT-ECG test set in $0.10$ seconds, over $3{,}270\times$ faster than MIX and $1{,}719\times$ faster than LIME at inference. The training cost is therefore amortized over the test set, and the per-instance saving grows with the number of instances explained. This separation also suits a centralized-training, distributed-deployment setting: a well-resourced institution trains the explainer once, after which it runs on CPU via a single forward pass at deployment sites without GPUs.

### D.3. Computational Cost: FLOPs, Parameters, and Memory

Table 27 reports FLOPs per explanation, explainer parameter count, and peak memory on Libras across the three backbones. Baselines query the target model repeatedly per explanation (IG along its integration path, LIME and KernelSHAP over perturbed samples), so their FLOPs scale with target-model complexity. XMA instead uses a single gradient query plus one forward pass of a fixed, compact explainer (267.5K parameters), independent of the target model's size.

The explainer is the same compact network (267.5K parameters) regardless of backbone, so XMA's FLOPs advantage grows with target-model complexity: from $19\times$ fewer FLOPs than IG on ResNet (209K parameters) to $47\times$ fewer on PatchTST (6.39M parameters). The added memory from the explainer is modest, about 1 MB across all backbones.

### D.4. Batch Inference on Multiple Instances

The per-instance figures in Section 4.3 are measured on a single instance on GPU. In deployment, many signals are explained together, so we also measure throughput on a batch of 200 MIT-ECG instances (Table 28).

XMA processes a batch of $B$ instances with a single forward and a single backward pass through $f_\theta$ (for the Input $\times$ Gradient features) followed by a single forward pass of the explainer, so it benefits directly from batching: at batch size 64 it explains all 200 instances in 0.23 seconds across four sequential batched passes, against 152.25 seconds for MIX, a $662\times$ reduction. Peak activation memory scales linearly with batch size for both XMA and the query-based baselines, so the per-instance memory characterized in Table 27 extrapolates directly; XMA's 267.5K explainer parameters remain fixed regardless of batch size.

These measurements are on CPU, reflecting resource-constrained deployment without a dedicated GPU.

*Table 19.* Faithfulness on Libras with varying evaluation step size $\delta$, training fixed at $\chi = 0.1$. Scores closely match Table 18.

| Architecture | $\delta = 0.05$ | $\delta = 0.1$ | $\delta = 0.2$ |
|---|---|---|---|
| FCN | 0.936 | 0.929 | 0.927 |
| PatchTST | 0.961 | 0.955 | 0.966 |
| ResNet | 0.923 | 0.934 | 0.932 |

*Table 20.* Faithfulness and robustness on Libras for explainers trained with Gaussian versus FGSM perturbations, evaluated under FGSM attacks and random noise. Best per architecture in bold.

| Architecture | Training | Faithfulness | Robustness (FGSM) | Robustness (Random) |
|---|---|---|---|---|
| FCN | FGSM | 0.932 | **0.783** | 0.818 |
| | Gaussian | **0.938** | 0.730 | **0.827** |
| PatchTST | FGSM | **0.964** | 0.599 | 0.608 |
| | Gaussian | 0.962 | **0.715** | **0.700** |
| ResNet | FGSM | 0.926 | **0.664** | 0.683 |
| | Gaussian | **0.946** | 0.617 | **0.707** |

*Table 21.* Effect of removing ground-truth important versus redundant features on MIT-ECG (2-class; random level: accuracy and confidence 0.50).

| Removed | Accuracy | F1 | Confidence |
|---|---|---|---|
| None (original) | 0.854 | 0.921 | 0.840 |
| Important (GT) | 0.440 | 0.611 | 0.441 |
| Redundant (non-GT) | 0.843 | 0.915 | 0.839 |

*Table 22.* Effect of removing ground-truth important versus redundant features on MCCE synthetic data (2-class; random level: accuracy 0.50).

| Removed | Accuracy | F1 |
|---|---|---|
| None (original) | 0.985 | — |
| Important (GT) | 0.499 | 0.333 |
| Redundant (non-GT) | 0.894 | 0.892 |

*Table 23.* Jaccard robustness on Libras (FCN) across binarization thresholds. XMA and IG occupy the top two positions at every threshold; the ranking does not reverse. Best per row in bold.

| Threshold | XMA | IG | LIME | KernelSHAP |
|---|---|---|---|---|
| 10% | 0.717 | **0.726** | 0.091 | 0.072 |
| 15% | **0.747** | 0.746 | 0.122 | 0.095 |
| 20% | **0.783** | 0.780 | 0.170 | 0.130 |
| 25% | **0.795** | 0.779 | 0.223 | 0.157 |
| 30% | **0.803** | 0.782 | 0.322 | 0.193 |

## E. Theoretical Analysis

In this section, we provide a formal proof for the **Faithfulness-Preserving Segmentation** mechanism proposed in Section 3.3. Our objective is to prove that the proposed stratified segmentation algorithm strictly preserves the faithfulness score when it is calculated with ratio step $\delta$. Specifically, we show that for any given step $i$, the set of features assigned to the $i$-th segment tier corresponds exactly to the features falling within the theoretical ranking range $[i \cdot \delta, (i+1) \cdot \delta]$ of the original attribution map $\mathcal{A}$, where $\delta$ is the ratio step (i.e. $step_r$ in Section 3.3). We define the score of $x \in \mathbf{x}$ on attribution $\mathcal{A}$ as $\mathcal{A}_x$.

*Table 24.* Reverse ablation on MIT-ECG. Removing distillation (Metric-Only) collapses all metrics; removing the metric loss (XM) is competitive but below the full model. Best per metric in bold.

| Variant | AUPRC | AUP | AUR |
|---|---|---|---|
| Full XMA | **0.890** | **0.594** | 0.877 |
| XM (distillation only) | 0.875 | 0.539 | **0.889** |
| Metric-Only (no distillation) | 0.429 | 0.330 | 0.577 |

*Table 25.* Reverse ablation on Libras: full model versus metric-only (MO), per backbone. Best per metric group in bold.

| Backbone | Faithfulness | | Robustness | |
|---|---|---|---|---|
| | Full | Metric-Only | Full | Metric-Only |
| FCN | **0.937** | 0.925 | **0.826** | 0.820 |
| PatchTST | **0.974** | 0.958 | 0.622 | **0.690** |
| ResNet | **0.945** | 0.917 | **0.714** | 0.699 |

*Table 26.* End-to-end cost of XMA on MIT-ECG (NVIDIA A5000), with baseline inference times for comparison. Stage 1 and Stage 2 are incurred once; inference is the cost over the full test set.

| Method | Stage | Time (s) |
|---|---|---|
| XMA | Stage 1: Expert generation | 693.50 |
| | Stage 1: Expert selection | 406.51 |
| | Stage 1: Gate training | 793.32 |
| | Stage 2: Explainer training | 354.95 |
| | **Total training** | **2248.28** |
| | **Inference (test set)** | **0.10** |
| MIX | Inference (test set) | 327.00 |
| LIME | Inference (test set) | 171.91 |
| TIMING | Inference (test set) | 3.03 |

*Table 27.* FLOPs per explanation, explainer parameters, and peak memory on Libras across three backbones. XMA uses a single query; baselines use 25–50. Target-model parameter counts are shown in parentheses.

| Backbone | Method | Queries | FLOPs | Params | Mem. (MB) |
|---|---|---|---|---|---|
| ResNet (209K) | IG | 50 | 378.7M | — | 0.81 |
| | LIME | 25 | 63.1M | — | 0.81 |
| | KernelSHAP | 25 | 63.1M | — | 0.81 |
| | XMA | 1 | **19.6M** | 267.5K | 1.83 |
| FCN (268K) | IG | 50 | 1330.6M | — | 1.03 |
| | LIME | 25 | 221.8M | — | 1.03 |
| | KernelSHAP | 25 | 221.8M | — | 1.03 |
| | XMA | 1 | **38.7M** | 267.5K | 2.05 |
| PatchTST (6.39M) | IG | 50 | 9476.4M | — | 34.15 |
| | LIME | 25 | 1579.4M | — | 34.15 |
| | KernelSHAP | 25 | 1579.4M | — | 34.15 |
| | XMA | 1 | **201.6M** | 267.5K | 35.17 |

**Problem Formulation**. Let $\mathcal{A} \in \mathbb{R}^{T \times d}$ be the continuous attribution map, where $N = T \times d$ is the total number of spatiotemporal features. Let $\Omega = \{1, \ldots, N\}$ be the set of all feature indices. We define a ranking function rank $: \Omega \to \{1, \ldots, N\}$ such that rank$(x)$ is the position of feature $x$ when $\mathcal{A}$ is sorted in descending order. Formally, if we define the sorted permutation $\pi$ such that $\mathcal{A}_{\pi(1)} \geq \mathcal{A}_{\pi(2)} \geq \cdots \geq \mathcal{A}_{\pi(N)}$, then rank$(\pi(k)) = k$.

*Table 28.* Inference time on 200 MIT-ECG instances. XMA is reported at batch sizes 1 and 64; speedup is relative to LIME. Lower time is better.

| Method | Total (s) | Per instance (ms) | Speedup vs. LIME |
|---|---|---|---|
| XMA (batch $= 64$) | **0.23** | **1.16** | $221.6\times$ |
| XMA (batch $= 1$) | 3.96 | 19.80 | $13.0\times$ |
| TIMING | 5.78 | 28.89 | $8.9\times$ |
| LIME | 51.62 | 258.11 | $1.0\times$ |
| MIX | 152.25 | 761.23 | $0.3\times$ |

Let $\delta \in (0, 1)$ be the ratio step (e.g., $\delta = 0.1$). The algorithm iterates through steps $i \in \{0, 1, \ldots, \frac{1}{\delta} - 1\}$. For each step, we verify that the generated segment set $\mathcal{S}_i$ satisfies the **Rank-Consistency Condition**.

**Segmentation Process**. We define the algorithm to build segments step-by-step. Let $R_k = k \cdot \delta$ be the cumulative ratio at step $k$. The algorithm defines the top-$k$ cutoff index as $C_k = \lfloor R_k \cdot N \rfloor$.

1. **Construct Top-K Sets:** The algorithm identifies the set of top-ranking features for the current upper bound $R_{i+1}$ and for each previous bound $R_0, R_1, \ldots, R_i$:

$$\mathcal{T}_{i+1} = \{x \in \Omega \mid \text{rank}(x) \leq C_{i+1}\} \tag{17}$$
$$\mathcal{T}_i = \{x \in \Omega \mid \text{rank}(x) \leq C_i\} \tag{18}$$
$$\ldots \tag{19}$$
$$\mathcal{T}_1 = \{x \in \Omega \mid \text{rank}(x) \leq C_1\} \tag{20}$$
$$\mathcal{T}_0 = \emptyset \tag{21}$$
$$\tag{22}$$

2. **Set Difference (Stratification):** The active set of features $\mathcal{S}_i$ for the current band is derived by the set difference:

$$\mathcal{S}_i = \mathcal{T}_{i+1} \setminus \left(\bigcup_{j \leq i} \mathcal{T}_i\right) \tag{23}$$

3. **Segment Merging within $\mathcal{S}_i$.** We partition the set of time indices $\mathcal{S}_i$ into a collection of disjoint, maximal contiguous segments, denoted as $\mathcal{S}\rceil\}_i$. Specifically, a subset of indices forms a segment in $\mathcal{S}\rceil\}_i$ if and only if the indices represent a sequence of consecutive integers that cannot be extended by adjacent integers within $\mathcal{S}_i$. For example, if $\mathcal{S}_i = \{1, 2, 5, 6, 7, 10\}$, the algorithm produces three distinct segments: $\{1, 2\}$, $\{5, 6, 7\}$, and $\{10\}$.

Since every segment is constructed by merging time points strictly contained within $\mathcal{S}_i$, the importance ranking of any point belonging to a segment $\in \mathcal{S}\rceil\}_i$ remains consistent in $\mathcal{S}_i$.

**Illustrative Example for Rank-Consistency Condition.** To clarify the segmentation mechanism, consider a feature space of 90 time points. We evaluate importance at incremental ratio steps. Let $\mathcal{T}_k$ denote the cumulative set of top features selected up to step $k$. We define the incremental set of features $\mathcal{S}_i$, specific to the interval between step $i$ and $i + 1$, as the difference between the current cumulative set and the union of all previous sets:

$$\mathcal{S}_i = \mathcal{T}_{i+1} \setminus \left(\bigcup_{j \leq i} \mathcal{T}_j\right) \tag{24}$$

Suppose the top $10\%$ ($0.0 - 0.1$ ratio) most important features are identified by descending attribution score. Then, $\mathcal{T}_1 = \{1, 2, 5, 8, 9, 10, 18, 19, 20\}$, and taking $\mathcal{T}_0 = \emptyset$, we have $\mathcal{S}_0 = \{1, 2, 5, 8, 9, 10, 18, 19, 20\}$. At the next threshold (Step 2, top $20\%$), the cumulative set is, for example, $\mathcal{T}_2 = \mathcal{T}_1 \cup \{6, 7, 22, 23, 24, 25, 26, 27, 28\}$. Applying our formula, we

derive the specific segments for this interval strictly from:

$$\mathcal{S}_1 = \mathcal{T}_2 \setminus \left( \bigcup_{j \leq 1} \mathcal{T}_j \right) = \{6, 7, 22, 23, 24, 25, 26, 27, 28\}$$

We then apply the segmentation algorithm, which merges consecutive time indices within each disjoint set:

- **Tier 1 (Top 10%):** Grouping $\mathcal{S}_0$ yields segments $\{(1, 2), (5), (8, 9, 10), (18, 19, 20)\}$.

- **Tier 2 (Next 10% to 20%):** Grouping $\mathcal{S}_1$ yields segments $\{(6, 7), (22, \ldots, 28)\}$.

By enforcing strict set exclusion ($\mathcal{S}_1 = \mathcal{T}_2 \setminus \left( \bigcup_{j \leq 1} \mathcal{T}_j \right)$) prior to segmentation, we guarantee that segments from different importance tiers remain disjoint and never "leak" information. This constraint ensures hierarchical consistency: the top features identified at any specific ratio (e.g., 0.1) are preserved as distinct structural units and are never merged with features from lower-importance groups. To strictly maintain this ranking in the final output, we compute a single score for each segment by averaging the attribution values of its constituent time points, and subsequently assign this uniform score to every time point within that segment.

**Proposition E.1** (Rank-Consistency of Stratified Segmentation). *Given a ratio step $\delta$, let $\mathcal{S}_i$ be the set of feature indices identified by the algorithm at step $i$ (representing the range from $i \cdot \delta$ to $(i+1) \cdot \delta$). For any feature $x \in \mathcal{S}_i$, the normalized rank of $x$ in the original attribution map satisfies:*

$$i \cdot \delta \leq \frac{rank(x)}{N} \leq (i+1) \cdot \delta \tag{25}$$

*This implies that the segmentation strictly preserves the importance hierarchy of the original attribution map $\mathcal{A}$ without leakage between tiers.*

*Proof.* According to the **Segmentation Process**, we have:

- **Cardinality Analysis:** Substituting the definitions of $\mathcal{T}_0, \mathcal{T}_1, \ldots, \mathcal{T}_{i+1}$:
$$\mathcal{S}_i = \{x \in \Omega \mid \text{rank}(x) \leq C_{i+1}\} \cap \{x \in \Omega \mid \text{rank}(x) \geq C_i\} \tag{26}$$

  This simplifies to:
$$\mathcal{S}_i = \{x \in \Omega \mid C_i \leq \text{rank}(x) \leq C_{i+1}\} \tag{27}$$

- **Normalization:** Given $x \in S_i$, dividing the inequality by the total count $N$, and substituting $C_k \approx k \cdot \delta \cdot N$:
$$\frac{i \cdot \delta \cdot N}{N} \leq \frac{\text{rank}(x)}{N} \leq \frac{(i+1) \cdot \delta \cdot N}{N} \tag{28}$$

$$i \cdot \delta \leq \frac{\text{rank}(x)}{N} \leq (i+1) \cdot \delta \tag{29}$$

**Conclusion:** Since the set operation $S_i = \mathcal{T}_{i+1} \setminus \left( \bigcup_{j \leq i} \mathcal{T}_i \right)$ is mathematically equivalent to selecting the disjoint rank interval $(C_i, C_{i+1}]$, the resulting segments in $\mathcal{S}_i$ are guaranteed to contain exclusively those features falling within the $i$-th importance tier of $\mathcal{A}$. Thus, the resulting segmentation map M keeps features strictly within the ranges Si with respect to the quantization step $\delta$. Intuitively, because we strictly enforce that segment merging (connectivity) occurs only between consecutive points within the specific rank band $[R_i, R_{i+1}]$, the topological structure of the features is preserved at each level. This design ensures that high-salience regions remain distinct and are never artificially diluted by merging with lower-salience background noise. $\square$

We proceed to demonstrate that assigning the average attribution score to all time points within a segment strictly preserves the composition of each feature group $\mathcal{S}_i$. Consequently, if we were to reconstruct the feature hierarchy based on these updated scores, the set of features constituting each $\mathcal{S}_i$ would remain identical to the original. **Illustrative Example for score consistency.** From the illustrative example for the Rank-Consistency Condition, we have:

- **Tier 1 (Top 10%):** Grouping $\mathcal{S}_0$ yields segments $\{(1,2),(5),(8,9,10),(18,19,20)\}$.

- **Tier 2 (Next 10% to 20%):** Grouping $\mathcal{S}_1$ yields segments $\{(6,7),(22,\ldots,28)\}$.

Now, since, in the attribution map, the importance score of the top 10% is always greater than or equal to that of the next 10%–20% (by the descending-and-select mechanism), $\min(A_1, A_2, A_5, A_8, A_9, A_{10}, A_{18}, A_{19}, A_{20}) \geq \max(A_6, A_7, A_{22}, \ldots, A_{28})$. We define two values, $\min(\mathcal{T}^{0.1})$ and $\max(\mathcal{T}^{0.1\to0.2})$, where $\mathcal{T}^{0.1}$ and $\mathcal{T}^{0.1\to0.2}$ are the two tier sets: $\mathcal{T}^{0.1} = \{A_1, A_2, A_5, A_8, A_9, A_{10}, A_{18}, A_{19}, A_{20}\}$, and $\mathcal{T}^{0.1\to0.2} = \{A_6, A_7, A_{22}, \ldots, A_{28}\}$. We have: $\min(\mathcal{T}^{0.1}) \geq \max(\mathcal{T}^{0.1\to0.2})$. After merging, the score of each point is the average of scores of all time points in its segment:

- $A_1, A_2$ are assigned as $(A_1 + A_2)/2 \geq \min(\mathcal{T}^{0.1})$

- $A_5$ is assigned as $A_5 \geq \min(\mathcal{T}^{0.1})$

- $A_8, A_9, A_{10}$ are assigned as $(A_8 + A_9 + A_{10})/3 \geq \min(\mathcal{T}^{0.1})$

- $A_{18}, A_{19}, A_{20}$ are assigned as $(A_{18} + A_{19} + A_{20})/3 \geq \min(\mathcal{T}^{0.1})$

Then, all scores of $\mathcal{T}^{0.1}$ are still $\geq \min(\mathcal{T}^{0.1})$.

For $\mathcal{T}^{0.1\to0.2}$:

- $A_6, A_7$ are assigned as $(A_6 + A_7)/2 \leq \max(\mathcal{T}^{0.1\to0.2})$

- $A_{22}, \ldots, A_{28}$ are assigned as $(A_{22} + \cdots + A_{28})/7 \leq \max(\mathcal{T}^{0.1\to0.2})$

Then, all scores of $\mathcal{T}^{0.1\to0.2}$ are still $\leq \max(\mathcal{T}^{0.1\to0.2})$. Overall, all scores of $\mathcal{T}^{0.1}$ are still $\geq \min(\mathcal{T}^{0.1})$, all scores of $\mathcal{T}^{0.1\to0.2}$ are still $\leq \max(\mathcal{T}^{0.1\to0.2})$ and $\min(\mathcal{T}^{0.1}) \geq \max(\mathcal{T}^{0.1\to0.2})$. It follows that all scores of features of $\mathcal{T}^{0.1}$ are still $\geq$ all scores of features of $\mathcal{T}^{0.1\to0.2}$, so that we still select $S_0$ then $S_1$ across the two tiers with the updated scores. It is easy to see this if we make an example with a lower tier, e.g., $\mathcal{T}^{0.1}$ vs. $\mathcal{T}^{0.5\to0.6}$ instead of consecutive tiers. We prove it formally below.

**Proposition E.2** (Feature Group Invariance). *Let $Seg_A \subseteq \mathcal{S}_i$ be a segment generated at step $i$, and $Seg_B \subseteq \mathcal{S}_j$ be a segment generated at a later step $j$ (where $j > i$, implying $Seg_B$ is of lower importance). Let $\mu(Seg)$ denote the score assigned to a segment, defined as the average of the attribution values of points within it:*

$$\mu(Seg) = \frac{1}{|Seg|} \sum_{x \in Seg} \mathcal{A}_x \tag{30}$$

*Then, the score of the higher-tier segment is greater than or equal to the score of the lower-tier segment:*

$$\mu(Seg_A) \geq \mu(Seg_B) \tag{31}$$

*Proof.* Let $\mathcal{A}_{\text{sorted}}$ be the attribution values sorted in descending order. Let $C_{i+1}$ be the rank cutoff index separating tier $i$ and the subsequent tiers. Let $\theta_i$ be the attribution value at this cutoff rank, i.e., $\theta_i = \mathcal{A}_{\text{sorted}}[C_{i+1}]$.

1. **Bounds of Tier $i$ (Higher Importance):** By the definition of the detailed algorithm in proposition E.1, every point $x \in \mathcal{S}_i$ (and thus every $x \in Seg_A$) satisfies the condition that its rank is better (lower) than or equal to $C_{i+1}$. Since all points of $Seg_A$ are in $\mathcal{S}_i$, we have:

$$\forall x \in Seg_A, \quad \mathcal{A}_x \geq \min_{u \in \mathcal{S}_i} A_u \tag{32}$$

Consequently, the average value $\mu(Seg_A)$ must also be bounded below by this minimum:

$$\mu(Seg_A) \geq \min_{u \in \mathcal{S}_i} A_u \tag{33}$$

2. **Bounds of Tier $j$ (Lower Importance):** For the subsequent tier $j$ (where $j > i$), the points belong to a rank range strictly worse (higher) than $C_{i+1}$. This implies that these points were *not* selected in step $i$, meaning their attribution values must be less than or equal to the minimum $\min_{u \in S_i} A_u$:

$$\forall y \in Seg_B, \quad \mathcal{A}_y \leq \max_{v \in S_j} A_v \leq \min_{u \in S_i} A_u \tag{34}$$

Consequently, the average value $\mu(Seg_B)$ must be bounded above by this threshold:

$$\mu(Seg_B) \leq \max_{v \in S_j}(\mathcal{A}_v) \leq \min_{u \in S_i} A_u \tag{35}$$

3. **Transitivity:** Combining the inequalities derived in Equations 33 and 35:

$$\mu(Seg_A) \geq \min_{u \in S_i} A_u \geq \max_{v \in S_j}(\mathcal{A}_v) \geq \mu(Seg_B) \tag{36}$$

**Conclusion:** We have proven that by averaging the scores within segments derived from stratified rank sets, the resulting segment scores strictly adhere to the original hierarchy. A segment generated in the Top 10% tier will mathematically always have a score higher than or equal to that of a segment generated in the Top 20% tier. This confirms that while the segmentation process reduces granularity via smoothing, it never inverts the ranking of importance, provided that the faithfulness evaluation employs deterministic tie-breaking, e.g., selecting features with equal scores by temporal order rather than random selection in the case of features with the same importance scores. $\qquad\square$

