# OpenReview forum: "Unified Time Series Explanations via Amortized Optimization and Instance-level Multi-Expert Knowledge Distillation"
_ICML.cc/2026/Conference — ICML 2026 regular_

### Official Review · Reviewer_KX16 · 2026-02-23

**Soundness:** 2
**Presentation:** 2
**Significance:** 3
**Originality:** 3
**Overall Recommendation:** 4
**Confidence:** 5

**Summary:**

The paper proposes XMA, a unified framework for explaining time-series classifiers that aims to combine the complementary strengths of existing post-hoc explainers with the efficiency of a learned explainer. The method generates multiple attribution maps per instance using a set of "expert" explainers and selects the instance-specific best explanation based on a metric that combines faithfulness and robustness (IMEKD). Then, it trains a neural explainer to distill the selected expert map while also optimizing auxiliary objectives intended to improve faithfulness and stability under perturbations (SAOE). Finally, it post-processes point-wise attribution maps into segment-based explanations using a rank-preserving grouping procedure that maintains the faithfulness score at specific sparsity levels (FPS)

**Compliance With Llm Reviewing Policy:**

Affirmed.

**Key Questions For Authors:**

- Using y-true can be incorrect when the model misclassifies and then the explainer is trained/selected to support a class the model did not choose. This can systematically distort "faithfulness" to the model's actual decision boundary. Why is the faithfulness objective tied to the true class rather than the model's predicted class (or a user-specified target)? How do you handle misclassified instances during IMEKD selection and SAOE training?
- In what precise sense is SAOE "semi-amortized"? Is there any test-time refinement/optimization, or is it purely amortized with auxiliary losses?
- The claim "there is typically one expert that optimally identifies critical features for each instance" is asserted but not theoretically justified. It is a hypothesis that needs careful empirical support and ablations. How often does the "best expert" differ? How sensitive is selection to metric choice?)
- IMEKD requires running all experts per training instance and evaluating faithfulness+robustness per expert (which itself requires repeated masking/forward passes and noise perturbations). The paper emphasizes shifting costs to training but does not quantify the training-phase cost of expert generation and per-instance selection (which can dominate total compute). What is the end-to-end computational cost of IMEKD (i.e., running all experts + evaluating faithfulness/robustness per instance)? How does this compare to (i) training the classifier, and (ii) using a strong optimization-based explainer at test time?
- Before MSE distillation, do you normalize expert maps (scale/sign, per-instance normalization, percentile normalization, etc.)? If not, why is MSE an appropriate objective for distillation across heterogeneous explainers?
- "We observe that while no single explanation method is universally superior across all data sample" is a well-known problem in XAI literature: explainers disagree. Add citations to disagreement/instability of explainers and/or benchmark studies. Clarify whether the "typically one optimal expert per instance" is empirical (and under which metric) rather than a general fact.
- What baseline signal is used for masking (and does it differ by dataset)? How sensitive are results to the baseline choice?
- Why Gaussian perturbations as the primary robustness model? Have you tested robustness under time-series-specific perturbations (time warping, jitter, missingness, amplitude scaling, resampling)?
- Many UEA datasets are multi-class, and ROC-AUC computation details (macro/micro/one-vs-rest) are not specified. For multi-class datasets, how is ROC-AUC computed under masking (macro/micro/one-vs-rest)? Are results statistically stable across seeds?
- Fig. 3 distinguishes batch sizes (1 vs 64). Where exactly is batching implemented, and how does batching affect time and memory across baselines?

**Limitations:**

I was excited to start reading this manuscript because time series and XAI are two areas I really enjoy researching and working on. The abstract is promising, and the proposal looks sound. However, I quickly became disappointed. The main limitation of this manuscript is language. I strongly recommend proofreading in English or (at a minimum) using a grammar checker like Grammarly, as the manuscript contains several grammatical issues that are unacceptable for a top-tier venue such as ICML. Some sections are difficult to understand, not because the mathematics is overly complex, but because there are so many language issues that it takes multiple readings to figure out what the text means.

Would be great to see proper citations to support many statements such as "However, the architectural complexity that drives this performance renders these models opaque "black boxes" to human users," "Consequently, XAI for time series has emerged as a vital area of research, aiming to bridge the gap between high-performance modeling and human interpretability," "The first is ante-hoc explanation, which focuses on ...," "The second perspective is post-hoc explanation, which aims to interpret a fixed, pre-trained...," Among many other statements/assumptions with no proper reference to XAI/TS literature. Authors do cite some adjacent claims (e.g., point-wise attribution literature), but specific high-level framings are largely uncited.

Consider increasing and standardizing the font sizes in Fig 1. Some words are really small and difficult to read. Fig. 1 is confusing. The legend says "The training objective combines three loss terms," but in the figure, we see four different losses in addition to an $L_{total}$ (the 5th different loss presented in Fig. 1). The arrow scheme lacks a coherent flow (multiple arrows converge and it is not immediately clear which losses are computed where), making it very hard to understand how all those losses are combined. The paragraph opening section 3 is not very straightforward. I was expecting Fig 1 to help, but it did not.

A lack of references is a recurring issue. Section 3 introduces many non-original concepts with almost no references. For example, did you create that "principle of sufficiency"? What is the theory supporting that "should be sufficient for the model to maintain high confidence in the true class"? This claim and the formulations related to the Faithfulness Loss have no value, with no references to validate them or provide theoretical proofs / empirical evaluations. Also, it targets the true class explicitly, which is a conceptual choice that should be justified (especially when the classifier misclassifies; then it is not "faithful to the model's decision").

Same issue with the Robustness Loss. The XAI literature includes many papers on robustness. However, the authors presented a robustness formalization out of nowhere. "To the best of our knowledge, this is the first work to directly incorporate robustness and faithfulness into a differentiable objective for training an explanation network". Even without external verification, this reads as an overreach: objective-based training of explainers with fidelity/stability regularizers is a known pattern in post-hoc explainer learning.

Section 3 needs an overall restructuring to improve the reading flow. There is a lack of connection among sequential concepts, and some "jump forward then back," as, for example, with the metric M in equation 2, which relies on concepts scattered across multiple sections not discussed up to equation 2. These jumps break the method comprehension.

Section 3: Not clear if SAOE is trained with pairs composed with the best attribution map for an instance or with several ones produced by IMEKD. From the text, I infer several maps, but I am not sure which ones (mathematically, it is a single teacher map per instance). The first paragraph of Section 3 really needs to be rewritten to ensure clarity.

3.1 and 3.2 rely on different optimization processes using a metric M. As far as I understand, M can differ in 3.1 and 3.2. If you are not using the exact same metric in equations 2 and 3, you should not use the same symbol because it hinders understanding. The processes are different, relying on different arguments and metrics. Furthermore, in 3.2, we see "that maximizes a specific evaluation metric," which can be faithfulness or robustness. This is not specific.

In summary, this paper addresses major gaps in TSC-XAI literature. However, the text requires major restructuring. I strongly encourage the authors to continue their valuable research and improve this work, but I am not sure there will be enough time to address such a large volume of content during the short rebuttal period.

**Strengths And Weaknesses:**

- Soundness. Combining multi-explainer generation with instance-wise selection and downstream distillation is a reasonable way to exploit disagreement between explainers while amortizing inference cost. Incorporating faithfulness and robustness losses during explainer training is directionally aligned with common evaluation criteria in XAI. However, the framework's faithfulness formulation appears tied to the true label rather than explicitly to the model's predicted decision. When the model misclassifies, optimizing "faithfulness" to the true label can produce explanations that are not faithful to the model's actual decision process, which is problematic for post-hoc explanation of a fixed classifier. The paper motivates semi-amortized optimization as involving refinement via objective evaluations, but the described SAOE stage reads primarily as amortized learning with additional regularization rather than a clear test-time refinement procedure. In my opinion, the term "semi-amortized" looks like a terminology stretch (it is closer to "amortized explainer with objective-based regularization"). Furthermore, the learned gating network used to approximate discrete masking can become a hidden failure mode (distribution shift between training signals and real attribution maps, sensitivity to architecture/hyperparameters). Robustness is defined primarily via small Gaussian perturbations and similarity of attribution maps, but I am not sure if this setup captures robustness to realistic time-series nuisance factors.

- Presentation. The overall pipeline components are conceptually structured, and definitions for time series / TSC are generally standard and understandable. However, frequent missing citations for broad claims and methodological assumptions weaken scientific grounding (details ahead). Mismatches between caption and diagram regarding how many losses are combined, small fonts, and an arrow scheme that does not clearly represent the computation graph/optimization flow. Language issues are frequent enough to be distracting for a top-tier venue.

- Significance. If rigorously validated, the approach could be practically useful. Fast inference-time explanations via a learned explainer, while leveraging multiple existing explainers during training, remains a gap in the TSC-XAI literature. However, I believe that the training-time cost of generating and scoring multiple expert explanations per instance could be very high, and the paper does not sufficiently quantify this total cost relative to the claimed efficiency benefits. Evaluation on multivariate datasets relies on a limited baseline set (largely generic explainers dating back to 2017), which weakens the strength of "SOTA" claims for time-series XAI broadly. The faithfulness metric used for some datasets may be weakly discriminative due to time-series redundancy.

- Originality. The combination of (i) instance-wise expert selection, (ii) distillation into a neural explainer, and (iii) segment post-processing with a rank-based invariance argument is indeed a coherent integration that is not commonly packaged as a single end-to-end framework. However, multiple components resemble established XAI patterns (teacher/student distillation, objective-regularized explainer learning, stability/perturbation-based robustness), so broad "first work" claims appear overstated.

General comments:

- I recommend a careful read of the abstract and introduction to streamline both texts and reduce redundant content. The current version is a bit wordy. Re-stating the same generalities should be avoided.

- There are LaTeX formatting issues in quotation marks. This issue is minor, but when reproduced across the whole text, it degrades readability. The correct LaTeX pattern is ``...content...''

- The reference "such as PatchTST (Nie et al., 2023) or TimesNet (Wu et al., 2022)" is repeated almost verbatim in two paragraphs of the intro. Avoid filler repetitions. Furthermore, there are many other well-known Time series transformers in the literature.

- Authors argue that IG-based methods require "expensive, iterative sampling" per instance. IG is typically deterministic numerical integration along a path with m steps (multiple forward/backward passes). It can be expensive, but describing it as "sampling" is technically sloppy and can mislead readers about what costs dominate.

- Minor: 2.1 Time series / TSC definitions are well-defined, although they lack some literature references. In "... posthoc explanation aims to interpret the decision-making ..." I suggest that authors replace the word interpret with explain. Although interpretability and explainability are sometimes used interchangeably, many authors prefer explainability because it is better aligned with post-hoc XAI methods.

- Minor: Check the need for a second pair of parentheses in the citation of Defs 2.1, 2.2, and 2.3.

- Minor: \emph{cf}. is the correct way to abbreviate the Latin word conferre (instead of the used c.f.). I further observe that conferre means "compare," which is not equivalent to "see."

- Some equations are long and exceed the column width. Consider [ some-x ; some-y ] to denote concatenations (making this symbology clear in the text). Also consider using \mathcal{L}_{\text{something}} to reduce label size.

- Gradient computation claim in 3.2. Computing gradients for an ML model requires (i) some kind of stochastic approximation, like the vanilla gradient proposed by K. Simonyan (Deep Inside Convolutional Networks: Visualising Image Classification Models and Saliency Maps), which I believe is not the case due to the "Second, it resolves ambiguity in stochastic models" observation; or (ii) white-box access to the gradient, and a backward pass, which is nontrivial in "real-time" settings and undermines the claim that inference is only a forward pass (unless using the Input-Only ablation variant). Furthermore, for standard deterministic TSC classifiers (the main setting), the "stochastic model ambiguity" argument is not central.

- The distillation loss is fragile without normalization. Attribution maps from different explainers often differ in scale, sign conventions, sparsity, and whether values are comparable across instances. MSE makes the student sensitive to these arbitrary scalings unless they normalize maps or use rank-based / distributional distillation. The main paper does not discuss normalization. The Appendix discusses normalization, but it applies only to section 3.3.

- The robustness metric is basically the RIS/ROS metrics introduced by Agarwal et al. in "OpenXAI: Towards a Transparent Evaluation of Post hoc Model Explanations" (no citation). However, without normalization, the formulation is less sensitive to data magnitudes or to the XAI methods. Furthermore, robustness is a reasonable regularizer, but it can also push the explainer toward low-variance, overly smooth maps that are stable but less sensitive to real decision-critical variations.

- Figure 3 says in its legend, "Note that XMA 1 and XMA 64 denote our method operating with batch sizes of 1 and 64, respectively." The surrounding efficiency paragraph does not discuss batch-size effects or why they are shown.

- For an ICML submission claiming SOTA broadly, the multivariate baseline set may be weak/incomplete (missing more recent multivariate TSC-specific post-hoc explainers, or at least stronger perturbation/optimization baselines).

---

> ### Author Rebuttal · Authors · 2026-03-31
>
> We are deeply grateful to the Reviewer for exceptionally thorough evaluation.
>
> >### Q1: True label?
>
> In TSC XAI, the standard AUC-based protocol inherently focuses on correctly predicted instances (i.e., implicitly assuming y_true = y_pred).
> Currently, no existing work systematically evaluates explanations on misclassified instances, though [A] uses both y(s) as complementary views.
> ||True|Pred|
> |-|-|-|
> |AUPRC|0.890|0.884|
> |AUP|0.594|0.570|
> |AUR|0.877|0.892|
>
> Our study on MIT-ECG above shows marginal performance changes of XMA when switching y_true and y_pred. Let the flip rate be % of flipped predicted labels when keeping only a subset of features.
> |Features|Flip Rate%|
> |-|-|
> |T|45.7%|
> |P|45.7|
> |T∩P|38.1 |
>
> On misclassified samples, let T & P be top-20% features from true- and predicted-label explanations. T∩P captures the most decision-critical features (38.1% flip rate).
>
> [A] A functional information perspective on model interpretation. 2022.
>
> >### Q2: Semi-amortized
>
> In Amos (2023), "semi-amortized" refers to iterative test-time refinement, which differs from XMA. The Input×Gradient at inference is an enriched input, not an optimization step, as confirmed by our Input-Only study (Sec. 4.4). XMA is a fully-amortized model with hybrid learning: distillation provides regression-based supervision & metric loss provides objective-based signal. **Objective-regularized amortized optimization** would be a more suitable term.
>
> >### Q3: Expert&metric
>
> The expert selection study on MIT-ECG shows no single expert dominates (**R. K5Da W2**).
> The remove-one-expert ablation (**R. K5Da Q3**) confirms the benefit of having diverse experts.
> ||FCN|PatchTST|ResNet|
> |-|-|-|-|
> |Faith. vs Composite|30.6%|27.8%|29.4%|
> |Robust. vs Composite|65.0%|55.6%|56.7%|
> |Faith. vs Robust.|12.8%|11.1%|10.6%|
>
> Metric sensitivity study on Libras above shows that faithfulness and robustness select the same expert for only ~11% of samples while our composite criteria are significantly better.
>
> >### Q4: Cost
>
> Please see **R. K5Da W1** for a runtime study and **R. BBky W4** for comparisons with recent SOTA explainers TIMING & Implet.
>
> >### Q5: Norm
>
> We apply per-instance min-max normalization to all attribution maps before distillation.
>
> >### Q6:  Explainer conflict
>
> We will add 3 proper citations: [B] formalize the disagreement problem; [D] ask "which explanation should I choose?" — our framework provides an automated, instance-level answer; and [C] document this for TSC specifically.
>
> [B] The Disagreement Problem in Explainable Machine Learning: A Practitioner's Perspective. 2022.
> [C] Robust explainer recommendation for time series classification. 2024.
> [D] Which explanation should I choose? A function approximation perspective to characterizing post hoc explanations. 2022.
>
> We clarify that "one optimal expert per instance" is empirical under our composite metric (**see Q3 above**).
>
> >### Q7: Baseline signal
>
> We use per-feature training mean as baseline (≈0 with Z-score norm), consistent with standard IG practice.
>
> |Baseline|AUPRC|AUP|AUR|
> |-|-|-|-|
> |Mean|0.890|0.594|0.877|
> |Gaussian|0.893|0.619|0.876|
> |Low-pass|0.889|0.598|0.876|
>
> The table above studies sensitivity on MIT-ECG (fixed model and varied evaluation baseline). Results are remarkably stable across all baselines.
>
> >### Q8: Perturb
>
> We study missingness and FGSM adversarial attacks beyond the chosen Gaussian perturbation in **R. K5Da Q5**. Regarding others: jitter is equivalent to Gaussian; amplitude scaling is absorbed by Z-score normalization; time warping shifts positions, making Jaccard inappropriate (planned as future work).
>
> |Model|Rob.(Gauss)|Rob.(Missing)|
> |-|-|-|
> |FCN|0.823|0.543|
> |PatchTST|0.697|0.561|
> |ResNet|0.707|0.551|
>
>
> The above table studies Gaussian-trained models under 20% missingness on Libras. Lower missingness robustness confirms it as a stronger perturbation than Gaussian noise — destroying the signal is more disruptive than corrupting it.
>
>
> >### Q9: ROC-AUC
>
> We use macro-averaged one-vs-rest ROC-AUC. We evaluate seed stability across 3 random seeds on Libras below.
>
> |Model|Faith.|Robust.|
> |-|-|-|
> |FCN|0.941 ± 0.001|0.822 ± 0.007|
> |PatchTST|0.970 ± 0.004|0.624 ± 0.013|
> |ResNet|0.939 ± 0.007|0.714 ± 0.015|
>
> Results are stable across seeds, with faithfulness std below 0.7% and robustness std below 1.5%.
>
> >### Q10: Batch
>
> Batching applies to the explainer's forward pass during XMA inference by processing a group of inputs concurrently.
>
> Baselines cannot benefit from batch processing due to their sequential nature. E.g., IG requires ~50 forward/backward passes and LIME/KernelSHAP require ~25 perturbed samples per instance.
>
> XMA requires one forward + one backward pass through f_θ (for Input×Gradient) plus one forward pass through the explainer for the entire batch. This is why the speed gap widens at batch=64.
>
> >### Others
>
> We will soften "SOTA" and "first work" claims.
>
> We have been addressing General Comments (GC1–GC13) and Limitations (L1-L8) thoroughly.

---

> > ### Author Rebuttal · Reviewer_KX16 · 2026-03-31
> >
> > I have no further questions.

---

> > > ### Author Response · Authors · 2026-04-01
> > >
> > > We sincerely thank the Reviewer for acknowledging that the concerns have been fully resolved. We are deeply grateful for the exceptionally detailed and rigorous review, which has substantially strengthened our manuscript. The questions regarding the true-label faithfulness formulation, the "semi-amortized" terminology, normalization, and robustness under diverse perturbation types, and many more, have led to important clarifications and new experiments that we believe add significant value to the paper. We will incorporate all the suggested revisions, including softened claims, added citations, improved figures, and restructured presentation, into the final version. Thank you again for the time and expertise invested in this review.

---

### Official Review · Reviewer_BBky · 2026-03-10

**Soundness:** 2
**Presentation:** 4
**Significance:** 3
**Originality:** 2
**Overall Recommendation:** 4
**Confidence:** 4

**Summary:**

This paper studies explainability for multivariate time series classification (MTSC), where existing post-hoc methods often suffer from high computational cost and inconsistent explanation quality. The authors propose XMA, a unified framework combining Multi-Expert Knowledge Distillation (IMEKD) and Semi-Amortized Optimization (SAOE) to improve efficiency and robustness of explanations. IMEKD distills knowledge from multiple expert explainers to train an explanation network, while SAOE refines explanations via a two-stage optimization process. Empirical results show effectiveness of the proposed method.

**Compliance With Llm Reviewing Policy:**

Affirmed.

**Final Justification:**

The rebuttal addressed my main concerns. I increased my rating.

**Key Questions For Authors:**

Please refer to the above weaknesses.

**Limitations:**

Yes

**Strengths And Weaknesses:**

**Strengths:**

- **S1**.	Presentation: the paper is well-structured and clearly written. The figures are easy-to-understand.
- **S2**.	Significance: the work addresses a well-known problem of high practical importance. The core idea of leveraging the “inconsistency” of existing XAI methods is a valuable contribution.
- **S3**.	Originality: the paper demonstrates originality by combining the amortized optimization and knowledge distillation into the SAOE framework for XAI.
- **S4**.	Soundness: the paper introduces a well-motivated and technically sound framework, named XMA, which addresses a genuine bottleneck in time series XAI.

**Weaknesses:**

- **W1**.	Significance: There are several highly related works: **TIMING** [a], **ORTE** [b], and **Implet** [c], which are also the time-series explainers. An additional work about **explainer recommendation for TSC** [d] might also be related to this submission. Without discussion on how the proposed method differs from the existing time-series explainers and experimental comparison with these recent methods, it is difficult to judge the novelty and unique contribution of this work.
- **W2**.	Soundness: the computational efficiency analysis is not sufficient. The authors are suggested to provide the results of MADD, FLOPs, parameter size, and running memory size.
- **W3**.	Soundness: the authors are suggested to discuss the metric’s sensitivity to the arbitrary threshold in the Jaccard Index on binarized maps (20% in the paper).
- **W4**.	Soundness: The paper evaluates on the SHAPEX synthetic suite, MIT-ECG, and 11 UEA multivariate datasets; for UEA it compares only with LIME, KernelSHAP, Input×Gradient, and IG. Why does this difference occur?
- **W5**.	Soundness: More ablations might be included to help improve the paper: the performance of single-expert distillation and the effect of expert pool size K.

[a] Hyeongwon Jang, Changhun Kim, and Eunho Yang. "Timing: Temporality-aware integrated gradients for time series explanation." in ICML 2025.

[b] Yue, Jinghang, et al. "Optimal information retention for time-series explanations." in ICML 2025.

[c] Meng, Fanyu, et al. "Implet: A Post-hoc Subsequence Explainer for Time Series Models." arXiv preprint arXiv:2505.08748 (2025).

[d] Nguyen, Thu Trang, Thach Le Nguyen, and Georgiana Ifrim. "Robust explainer recommendation for time series classification." Data Mining and Knowledge Discovery 38.6 (2024): pp. 3372-3413.

---

> ### Author Rebuttal · Authors · 2026-03-31
>
> We sincerely thank Reviewer BBky for the constructive feedback & for recognizing that our presentation is "well-structured and clearly written", the work "addresses a well-known problem of high practical importance", and our approach "demonstrates originality by combining amortized optimization and knowledge distillation".
>
> >### W1
>
> TIMING [A] enhances IG with temporality-aware stochastic baselines and segment-based masking. ORTE [B] learns a binary mask via an Adaptive Mask Generator with a novel Straight-Through Estimator and contrastive loss. Implet [C] provides explanations via importance subsequences. They all aim to improve explanation quality from a single methodological perspective. Rather than developing new attribution methods, XMA asks an entirely different question: "Can combining heterogeneous explanation methods in an instance-adaptive way produce explanations that exceed any individual method?" It is designed to be complementary to all these methods as experts in its pool.  If TIMING/ORTE/Implet outperform weaker methods, IMEKD will select them; if they underperform on others, alternative experts will compensate. This extensibility is a key feature of XMA: as stronger explainers emerge, they can be incorporated without modifying the framework.
>
> AMEE [D] aims to recommend the best explainer per dataset, sharing our observation that explainers disagree and no single method universally dominates. However, it operates at the dataset level rather than the instance level like XMA. Furthermore, AMEE acts as an evaluator and recommender rather than an explainer; users must still run the recommended method at test time, while XMA directly generates explanations.
>
> The table below compares TIMING, Implet & XMA on MIT-ECG (ORTE GitHub is incomplete, preventing reproduction).
>
> ||AUPRC|AUP|AUR|
> |-|-|-|-|
> |XMA|*0.890*|0.594|*0.877*|
> |Implet|0.552|0.429|0.853|
> |TIMING|0.468|*0.817*|0.096|
>
> XMA has substantially higher AUPRC than TIMING & Implet. TIMING has the highest AUP. However, its 0.096 AUR means it misses over 90% of the cardiologist-annotated important regions. In a clinical setting, an explainer that overlooks the vast majority of diagnostically relevant features would be insufficient for decision support. XMA has a far better precision-recall balance (AUP: 0.594, AUR: 0.877).
>
> >### W2
>
> The table below reports FLOPs per explanation, parameter count, and memory on Libras over 3 DNNs.
>
> |DNN|Method|Queries|FLOPs/Expl.|Params|Memory (MB)|
> |-|-|-|-|-|-|
> |ResNet (209K)|IG|50|378.7M|-|0.81|
> ||LIME|25|63.1M|-|0.81|
> ||KernelSHAP|25|63.1M|-|0.81|
> ||XMA|1|19.6M|267.5K|1.83|
> |FCN (268K)|IG|50|1,330.6M|-|1.03|
> ||LIME|25|221.8M|-|1.03|
> ||KernelSHAP|25|221.8M|-|1.03|
> ||XMA|1|38.7M|267.5K|2.05|
> |PatchTST (6.39M)|IG|50|9,476.4M|-|34.15|
> ||LIME|25|1,579.4M|-|34.15|
> ||KernelSHAP|25|1,579.4M|-|34.15|
> ||XMA|1|201.6M|267.5K|35.17|
>
> Our XMA explainer is a fixed compact FCN with 267.5K params. For FLOPs,  all baselines (e.g., IG) must query the target model 25–50 times per explanation, so their FLOPs scale linearly with target model complexity. However, XMA requires only a single query (for Input×Gradient) plus one forward pass through the explainer. Thus, its FLOPs advantage increases from 19× on ResNet (209K params) to 47× on PatchTST (6.39M params), highlighting the computational benefit of our amortized explanation. Memory overhead from the explainer is modest (~1 MB).
>
> >### W3
>
> The 20% threshold is commonly used in recent TSC XAI evaluations e.g., TIMING [A]. As suggested, the below Table reports Jaccard robustness vs multiple thresholds on Libras (FCN).
>
> |Threshold%|XMA|IG|LIME|KernelSHAP|
> |-|-|-|-|-|
> |10|0.717|0.726|0.091|0.072|
> |15|0.747|0.746|0.122|0.095|
> |20|0.783|0.780|0.170|0.130|
> |25|0.795|0.779|0.223|0.157|
> |30|0.803|0.782|0.322|0.193|
>
> XMA consistently dominates others. Importantly, no threshold reverses the overall method ranking, confirming that our results are not an artifact of the specific 20% binarization choice.
>
> >### W4
>
> The table below compares XMA with TIMING & Implet on UEA data Libras.
>
> |Model||Faith.|||Robust.||
> |-|-|-|-|-|-|-|
> ||XMA|TIMING|Implet|XMA|TIMING|Implet|
> |FCN|0.937|0.925|0.922|0.826|0.723|0.776|
> |PatchTST|0.974|0.887|0.886|0.622|0.482|0.759|
> |ResNet|0.946|0.896|0.878|0.714|0.588|0.595|
>
> XMA achieves the highest faithfulness on all models, while Implet has higher robustness on PatchTST.
>
> >### W5
>
> We show the effect of pool size K on Libras (FCN).
>
> |K|Faith. (AUC)|Robust. (Jaccard)|
> |-|-|-|
> |K=1(single best expert)|0.926|0.695|
> |K=2|0.939|0.820|
> |K=3(full pool)|0.942|0.817|
>
> K=1 achieves 0.926 faithfulness, but K=3 improves it further. Moving K from 1 to 2 increases Jaccard considerably. This suggests that expert diversity provides complementary supervision that significantly strengthens attribution stability. Adding the third expert further improves faithfulness while maintaining robustness, confirming the value of our IMEKD over committing to any single method.

---

> > ### Author Rebuttal · Reviewer_BBky · 2026-04-01
> >
> > Thanks for the detailed response. I have increased my rating.

---

> > > ### Author Response · Authors · 2026-04-01
> > >
> > > We sincerely thank the Reviewer for acknowledging that the concerns have been fully resolved and for increasing the rating. We are grateful for the thoughtful and constructive review, which motivated several important additions to our work, including comparisons with recent methods (TIMING, Implet), detailed computational efficiency analysis (FLOPs, parameters, memory), threshold sensitivity studies, and the expert pool size ablation. These contributions have meaningfully strengthened both the experimental rigor and the clarity of our paper. We will incorporate all the suggested revisions into the paper. Thank you again for the valuable feedback.

---

### Official Review · Reviewer_K5Da · 2026-03-12

**Soundness:** 2
**Presentation:** 3
**Significance:** 2
**Originality:** 3
**Overall Recommendation:** 4
**Confidence:** 5

**Summary:**

The authors propose a two-stage XMA framework. In the first stage, they use instance-level multi-expert knowledge distillation to generate attribution maps from a pool of XAI experts (e.g., LIME, KernelSHAP, IG) and select the optimal map per instance based on faithfulness and robustness metrics. In the second stage, they use a DNN explainer to predict the instance-optimal attribution map.

**Compliance With Llm Reviewing Policy:**

Affirmed.

**Final Justification:**

I am raising my score based on the rebuttal, but I think the paper needs significant improvement.

**Key Questions For Authors:**

1. Figure 3 is unclear to me. The proposed framework needs to generate multiple attribution maps for each instance by different XAI methods in a pool, what is the runtime of stage 1?

2. How does explanation quality degrade when the optimal attribution map for an instance is not well-approximated by any expert in the pool?

3. Have the authors considered evaluating on removing one expert during training and evaluating on instances where that expert would have been selected?

4. The faithfulness step size is fixed at 0.1. What is the sensitivity of the faithfulness metric to this value?

5. Have the authors considered replacing or augmenting the Gaussian robustness loss with adversarial perturbations (e.g., FGSM or PGD applied to the explainer)? If so, what tradeoffs in training complexity were observed? If not, can the authors provide  evidence that Gaussian noise is a sufficient robustness proxy for the time series domains tested?

**Limitations:**

The authors does not discuss the limitations of their work. Please see the comments regarding **Weaknesses**.

**Strengths And Weaknesses:**

**Strengths**
1. The core motivation of the work is that no single XAI method universally dominates, and that a learned amortized explainer can approximate instance-optimal explanations. It is well-motivated in my view and practically relevant for time series explanations, a domain where XAI methods remain underdeveloped compared to CV and NLP.

2. Rethinking XAI as an instance-level optimization problem rather than committing to a single method globally is practically important, given that instance may be quite different within the dataset.

**Weaknesses**
1. In stage 1, the framework requires running multiple point-wise post-hoc XAI methods on every training instance. For SHAP-based methods such as KernelSHAP which is expensive in the worst case with respect to the number of time steps, this overhead can be prohibitive even for moderate-length time series. The authors should report the stage 1 overhead and the time saving from stage 2. Without such an analysis, it is unclear whether the framework is efficient in practice.

2. In my view, the framework implicitly assumes that the ground-truth optimal attribution maps lies within the XAI expert pool. However, there is no theoretical or empirical justification for this assumption. At least, the authors should discuss criteria for selecting which experts to include, how performance degrades as pool quality deteriorates, the marginal benefit of adding new experts relative to their additional computational cost, etc.

3. The faithfulness loss is defined using a pre-defined masking step size (default is 0.1 in the paper). This hyperparameter directly determines which proportion of features and time steps is deemed “important” for the faithfulness evaluation. However, the fraction of truly important features and time steps is a dataset- and instance-level property that cannot be known a priori. A fixed global step size imposes an implicit hard assumption about explanation sparsity that is unlikely to hold uniformly across datasets, even across individual instances within a dataset. The authors provide no adaptive mechanism to set it and no theoretical argument for why 0.1 is a principled choice.

4. The robustness loss is computed by adding Gaussian noise to the input and penalizing attribution map divergence. From my perspective, adversarial perturbations, which find the minimal input change that maximally distorts the explanation, are a far more rigorous test of robustness. The use of Gaussian noise may cause the framework to overestimate robustness.

5. The authors conflates faithfulness that masking important features identified by an explanation degrade model performance, with a broader notion of fidelity to human intuition. But, a model may rely on  features that are statistically predictive but semantically meaningless. An explanation that is highly faithful to model behavior may therefore be different from a human interpretability standpoint.

---

> ### Author Rebuttal · Authors · 2026-03-31
>
> We sincerely thank the reviewer for the constructive feedback & for recognizing that our work is "well-motivated" and "practically relevant for time series explanations", particularly the observation that "rethinking XAI as an instance-level optimization problem is practically important."
>
> >### W1 Q1
>
> We agree with the reviewer. Below are the full costs on MIT-ECG (GPU).
> |Method|Stage|Time(s)|
> |-|-|-|
> |*XMA*|1: Expert Generation|693.5|
> ||1: Expert Selection|406.5|
> ||1: Gate Train|793.3|
> ||2: Explainer Train|354.9|
> ||*Sum*|*2248.2*|
> ||*Test*|*0.1*|
> |MIX|-|327.0|
> |LIME|-|171.9|
> |TIMING|-|3.0|
>
> XMA's design philosophy explicitly shifts computational cost from inference to training. The total training time (~37 mins) is a one-time cost. Once trained, the explainer generates explanations for the entire test set in 0.10 seconds (3,270× faster than MIX). For a hospital monitoring thousands of patients daily, the one-time cost becomes negligible compared to the cumulative inference savings.
>
> >### W2 Q2
>
> Thanks for this important point. We realized that our notation in Fig. 1 may cause misunderstanding that A* represents a globally optimal attribution map. A* denotes the best attribution map among the experts in the pool for a given instance.
>
> To see if the pool provides meaningful diversity, we report the selection rate for each expert on MIT-ECG. No single expert dominates, and rates range from 14.94% to 30.62%, confirming that different instances genuinely benefit from different expert configurations.
>
> We agree that this does not eliminate the theoretical limitation, and we will add discussion of pool selection criteria, the effect of pool quality, and guidelines for expert pool construction in revision.
>
> >### W3 Q4
>
> We train XMA with different masking step sizes χ and report faithfulness on Libras. We also vary χ at evaluation time (while keeping training fixed at 0.1) to disentangle the effect of training vs. evaluation granularity.
>
> *Table R1: Faithfulness for training (top) and testing (bottom)*
> |Model/χ|0.05|0.1|0.2|
> |-|-|-|-|
> |FCN|0.936|0.929|0.927|
> |PatchTST|0.961|0.954|0.965|
> |ResNet|0.923|0.934|0.932|
> |||||
> |FCN|0.936|0.929|0.927|
> |PatchTST|0.961|0.955|0.966|
> |ResNet|0.923|0.934|0.932|
>
> Results are stable across all settings, with variations consistently below 1% for each model. No single step size uniformly dominates others. These results indicate that XMA does not overfit to a specific masking value.
>
> We agree that an adaptive, instance-level step size mechanism could improve performance further, and will discuss this promising direction in the paper.
>
> >### Q3
>
> We conduct the suggested experiment on Libras under two settings: removing IG and removing Input×Gradient.
>
> *Table R2: Faithfulness (1: full, 2: remove)*
> |Model|V|IG|Input×Gradient|
> |-|-|-|-|
> |FCN|1|0.673|0.596|
> ||2|0.662|0.586|
> |PatchTST|1|0.804|0.803|
> ||2|0.797|0.798|
> |ResNet|1|*0.713*|*0.685*|
> ||2|0.718|0.691|
>
> The full pool has higher faithfulness than the reduced pool in 4/6 cases across both settings, confirming that having access to diverse experts during training provides higher-quality supervision. The modest degradation (< 1–3%) shows that remaining experts and the metric loss provide sufficient signal for the student to learn reasonable attributions even without the best teacher.
>
> >### W4 Q5
>
> The suggested idea of adversarial robustness for explanations, perturbing inputs to maximally distort attribution maps, as far as we know, has not been systematically studied in TSC XAI, and thus is highly interesting. The high cost of adversarial training particularly suits XMA because of its efficiency and its explainer network.
>
> *Table R3: We train two XMA variants on Libras, one with our Gaussian robustness loss (1) and one with FGSM adversarial perturbations (2), and evaluate them under FGSM attacks and Random noise.*
> |Model|V|Faith.|Rob.(FGSM)|Rob.(Gauss)|
> |-|-|-|-|-|
> |FCN|2|0.932|0.783|0.818|
> ||1|0.938|0.730|0.827|
> |PatchTST|2|0.964|0.599|0.608|
> ||1|0.962|0.715|0.700|
> |ResNet|2|0.926|0.664|0.683|
> ||1|0.946|0.617|0.707|
>
> The results reveal nuanced findings: (1) Gaussian training shows relative stability under FGSM and (2) FGSM training improves adversarial robustness on FCN & ResNet but not on PatchTST, suggesting a non-trivial trade-off between adversarial robustness & faithfulness that deserves further study.
>
> >### W5
>
> To fully understand this, we use MIT-ECG. If the model relies on the same ground truth (GT) features that human experts consider meaningful, removing them should degrade performance to near-random.
>
> |-|Acc|F1|Confidence|
> |-|-|-|-|
> |Original|0.854|0.921|0.840|
> |Remove GT|0.440|0.611|0.441|
> |Remove Other|0.843|0.915|0.839|
>
> Removing GT features drops both accuracy/confidence to the random-guess level, while removing other features only slightly affects results.
>
> However, we acknowledge that this alignment may not hold universally, as the reviewer pointed out. We will add this in the paper.

---

> > ### Author Rebuttal · Reviewer_K5Da · 2026-04-03
> >
> > Thank you for your rebuttal.
> >
> > 1. Your response to **W2 Q2** is a bit unclear to me.
> > 2. Regarding your response to **Q3**, the results of ResNet is somehow unexpected. Also, might be good to see the difference of removing LIME or KernelSHAP instead of removing gradient-based XAI methods.

---

> > > ### Author Response · Authors · 2026-04-04
> > >
> > > We sincerely thank the reviewer for carefully reading our rebuttal and for acknowledging that most concerns have been partially resolved. We appreciate that there are only two remaining points concerning our work.
> > >
> > > >### **R1**: W2/Q2 clarification
> > >
> > > We apologize for the unclear response. To clarify: we do not assume the globally optimal attribution lies within the pool. A* denotes the instance-best among available experts, not the theoretical optimum.
> > >
> > > The reviewer raised 3 specific sub-questions for W2 below:
> > >
> > > (1) Criteria for selecting experts: Our design principle is to maximize methodological diversity, we include experts with different computational paradigms (perturbation-based: LIME/KernelSHAP or gradient-based: IG/Input×Gradient). Once included, each expert is scored per instance using the composite metric M (Sec. 3.1, Eq. 2), combining faithfulness and robustness, and the highest-scoring expert is selected as teacher for that instance.
> > >
> > > (2) Marginal benefit vs. cost: Pool size ablation (**R. BBky W5**) shows K=1→K=2 yields the largest gain (+1.3% faithfulness, +12.5% robustness), with diminishing returns for K=3. Remove-one-expert results (**see R2 below**) show <4% degradation. A small diverse pool (K=3 to 5) captures most benefits without excessive overhead.
> > >
> > > (3) Pool quality degradation: When experts are removed, the metric loss compensates so that performance remains stable. However, metric loss alone produces poor explanations (AUPRC: 0.429, **R. ZLM3 R1**). The expert pool prevents blind optimization, while the metric loss prevents the student from being bounded by pool quality, together they are robust to pool imperfections.
> > >
> > > For Q2, based on results in **R2** below, we observe that XMA’s performance drops marginally, e.g., when we remove the best expert IG for FCN in **Q3** only reduce 0.011 in faithfulness, and with ResNet, metric loss can even have comparable performance with full pool.
> > >
> > > Expert selection shows diversity across datasets:
> > >
> > > MIT-ECG (s=step):
> > > |Expert|Select(%)|
> > > |-|-|
> > > |MIX(ws=24,s=9)|30.62|
> > > |MIX(ws=24,s=6)|20.84|
> > > |MIX(ws=36,s=9)|17.77|
> > > |MIX(ws=36,s=6)|15.84|
> > > |MIX(ws=28,s=6)|14.94|
> > >
> > > Libras (FCN):
> > > |Expert|%|
> > > |-|-|
> > > |LIME|49.4|
> > > |IG|9.4|
> > > |KeystoneIG variants |35.6|
> > > |Input×Gradient|3.9|
> > > |KernelSHAP|1.7|
> > >
> > > The distributions differ substantially, MIT-ECG shows balanced selection while Libras favors LIME.
> > >
> > > >### **R2** Q3 clarification
> > >
> > > Regarding the ResNet result (0.718 vs 0.713 when removing IG), which is only a 0.7% difference. This can occur because our system does not rely on distillation alone, the metric loss can occasionally refine beyond the teacher, especially when freed from a suboptimal one. The broader evidence is clear: combining weak explainers surpasses strong individual methods, XMA > TIMING & Implet (**R. BBky W4**), and K=3 > K=2 > K=1 (**R. BBky W5**).
> > >
> > > Nonetheless, as requested, we provide LIME and KernelSHAP removal results on Libras. Each table evaluates faithfulness only on instances where the removed expert would have been selected (results for w/o IG and w/o Input×Gradient are in the original rebuttal):
> > >
> > > w/o LIME:
> > > |Arch |w/o LIME|Full pool|
> > > |-|-|-|
> > > |FCN|0.678|0.706|
> > > |PatchTST|0.673|0.668|
> > > |ResNet |0.529|0.548|
> > >
> > > w/o KernelSHAP:
> > > |Arch|w/o KernelSHAP|Full pool|
> > > |-|-|-|
> > > |FCN|0.485 |0.445|
> > > |PatchTST|0.709|0.745|
> > > |ResNet |0.743|0.75|
> > >
> > > The full pool outperforms in most cases. The FCN w/o KernelSHAP improvement (+4%) is on a tiny subset, KernelSHAP accounts for only 1.7% of Libras instances, where the metric loss can more easily be refined from a lower starting point. We are grateful to the reviewer for pushing us to examine this, as it reveals an interesting property: removing a rare outlier expert may occasionally help because the remaining teachers provide more coherent, methodologically similar supervision, resulting in a warmer start that is easier for the metric loss to refine. Similarly, occasional small improvements when removing other experts (ResNet w/o IG, PatchTST w/o LIME) suggest that when the teacher is weaker, the metric loss has more room to refine beyond it. We do not assume more experts necessarily yield better explanations. Our practical approach is to collect diverse well-established explainers and let the metric loss compensate for pool imperfections. We believe this pragmatic design, which already produces significant improvements over strong individual explainers (see **R. BBky W4**), is a reasonable trade-off that we will discuss more thoroughly in revision.
> > >
> > > Crucially, metric loss alone without any teacher produces poor explanations (**see R. ZLM3 R1: Soundness**), confirming that these small improvements depend on remaining experts still providing a meaningful warm start. The expert pool prevents blind optimization, while the metric loss prevents the student from being bounded by pool quality. Together they enable combining weak generic explainers to surpass stronger specialized methods, which is the central contribution of this work.

---

### Official Review · Reviewer_ZLM3 · 2026-03-13

**Soundness:** 3
**Presentation:** 3
**Significance:** 2
**Originality:** 2
**Overall Recommendation:** 4
**Confidence:** 4

**Summary:**

This paper proposes a framework called XMA (eXplanation via Multi-expert and Amortized optimization) to address the computational inefficiency and inconsistent quality of current XAI methods for Time Series Classification. It first selects the "best" explanation for each instance by choosing the explanation with the best faithfulness scores, and this is then distilled into an "explanation model" which essentially performs amortized optimization.  This model is additionally optimized for faithfulness and robustness of the resulting attribution maps. The paper evaluates its proposed framework across 4 synthetic and 11 realistic time series datasets, demonstrating state of the art faithfulness, robustness, and computational speed.

**Compliance With Llm Reviewing Policy:**

Affirmed.

**Final Justification:**

The authors' rebuttal has clarified the main concerns I raised in my review; hence I am raising my score.

Concerns regarding lack of grounded uses cases, human-in-the-loop validation and overall significance still remain; but I understand that such methodological issues persist in the entire TSC XAI community and are not specific to this work. For this reason, my recommendation is a borderline accept.

**Key Questions For Authors:**

I wonder whether the authors can comment on my question regarding the missing critical ablation? That would help address the necessity of the explanation selection step.

**Limitations:**

The paper does not discuss any technical limitations of their work.

**Strengths And Weaknesses:**

**Soundness**: The arguments and experiments in the paper are generally sound. The claims are well-supported by experiments, and ablation experiments are also provided.
- One critical missing ablation is the following: the explainer model is trained using two losses, a distillation loss and a metric loss. While there is an ablation evaluating the impact of removing the distillation loss, and keeping only the metric loss, essentially bypassing the explanation selection mechanism completely.

**Presentation**: The paper is easy to read, and relevant related works are well-discussed.

**Significance**: This paper scores low on significance, for the following reasons:
- One of the main motivations of the paper is computational efficiency. While technically the paper is correct in that it proposes a faster method, the need for a faster XAI method is unclear, especially when existing methods are already somewhat fast. From Figure 3, the baselines take 0.1 seconds to produce an explanation, which already seems quite fast. What are real-world settings that demand fast explanation generation, where 0.1s is unacceptable? Overall I believe this motivation for fast explanation methods is weak.
- While the proposed method performs better on the quality metrics, it is unclear whether the difference in numerical quality metrics makes a difference for real world usage. For instance, are there any downstream human-in-the-loop use cases where humans prefer XMA explanations consistently to others? Do XMA explanations provide any additional insights that are missed by other methods? This is important to demonstrate the significance of the proposed approach.

**Originality**: The methods in the paper are mildly original. All the specific ingredients, such as choosing the best explanation (e.g.: MIX (Tran et al., 2025a)), and amortized explanation optimization (e.g.: Dynamask (Crabbé & Van Der Schaar, 2021)) are already explored in previous papers, and the originality in this paper is combining these ideas. In general, I don't find there to be any novel insight provided by this paper regarding time series explanations, or explanations in general; that haven't already been provided by other papers.

---

> ### Author Rebuttal · Authors · 2026-03-31
>
> We sincerely thank the reviewer for the constructive feedback & for recognizing that "the arguments and experiments are generally sound" and "the paper is easy to read."
>
> >### R1: Soundness
>
> As suggested, we conduct the reverse ablation: removing distillation entirely and retaining only the metric loss.
>
> Table R1: MIT-ECG
> |-|AUPRC|AUP|AUR|
> |-|-|-|-|
> |Full XMA (λ_mse=1,λ_faith=0.1,λ_rob=0.1) |0.890|0.594|0.877|
> |MSE-Only/XM (1,0,0)|0.875|0.539|0.889|
> |Metric-Only (0,0.1,1)|0.429|0.330|0.577|
>
> Table R2: Libras (F: Faith., R: Robust., MO: Metric-Only, Fu: Full)
> |-|F.Fu|F.MO|R.Fu|R.MO|
> |-|-|-|-|-|
> |FCN|0.937|0.925|0.826|0.820|
> |PatchTST|0.974|0.958|0.622|0.690|
> |ResNet|0.945|0.917|0.714|0.699|
>
> On MIT-ECG, without expert distillation, explainers must discover meaningful attributions from scratch using only the faithfulness/robustness gradient, a weak learning signal with poor local minima, thus leading to worse performance. MSE-Only (XM) performs reasonably but cannot exceed XMA. On Libras, Full XMA consistently achieves the best results. An exception on PatchTST robustness suggests that distillation can occasionally constrain robustness optimization. Overall, these results show that both components are complementary: expert distillation provides a high-quality warm start, while the metric loss refines the explanation beyond teacher quality.
>
> >### R2: Runtime
>
> We appreciate this thoughtful observation. However, faster XAI is well motivated in safety-critical domains such as ECG monitoring.
> E.g., [A] highlights the vital roles of XAI for ethical clinical decision-making and identifies computational expense as a key limitation. According to [B], the risk of death increases 5–10% per minute of delay for patients suffering ventricular fibrillation, while defibrillation is delayed beyond 2 minutes in 30% of cases. According to [C], healthcare data requires efficient batch processing, meaning per-instance XAI cost accumulates when monitoring many patients simultaneously.
> In developing countries, where patient volumes are high but computational resources are limited, the runtime becomes particularly critical.
>
> Under a batch of 200 MIT-ECG patients, it takes MIX, TIMING & LIME 152.3, 5.8 & 51.6 secs (CPU), resp. Thus, with MIX, the mortality risk increases significantly, while we have instant responses of 0.23 secs with XMA (660× faster). This time gap determines whether XAI-assisted monitoring can be deployed in resource-constrained and intense usage settings.
>
> We wish to clarify that computational efficiency is only one side of XMA; another motivation is explanation quality through multi-expert distillation, as confirmed by our ablation above.
>
> [A] 10.1016/j.bspc.2025.109325
>
> [B] 10.1097/CCM.0000000000000923
>
> [C] 10.1371/journal.pone.0279305
>
> >### R3: Human
>
> The highlighted evaluation gap applies broadly to the TSC XAI field, not only our paper. As far as we know, no existing TSC XAI method includes a human-in-the-loop evaluation. The field currently relies on ground truth alignment metrics and faithfulness proxies.
>
> Our MIT-ECG experiment provides human-grounded assessment with annotated ground truth from cardiologists. XMA improvement over MIX (Fig. 2) and qualitative visualizations (Fig. 5-9) show that our explanations align better with what domain experts identified as relevant regions.
>
> >### R4: Originality
>
> We agree that individual ingredients have precedents. We will soften our "first work" claims in revision. However, we respectfully clarify that the cited works differ substantially from XMA.
>
> MIX selects the best view among different Wavelet views of an input via an IG-based method. In contrast, XMA selects among fundamentally different XAI methods (with different computational paradigms and theoretical foundations) the most suitable one to explain an instance. At test time, MIX performs full iterative IG computation, while XMA produces explanations via a backward pass and single forward pass.
>
> Similarly, Dynamask fits a perturbation mask from scratch for every test instance via iterative gradient-based optimization. It is precisely the traditional expensive per-instance solver that XMA's amortized design replaces.
>
> Our contribution lies in the system-level integration: showing that instance-level multi-expert selection, combined with objective-regularized amortized training and faithfulness-preserving segmentation, produces explanations that exceed the quality of any individual expert. This "student exceeding teacher" result is non-obvious, since standard distillation is bounded by teacher quality. The metric loss enables this transcendence, and the multi-expert pool ensures diverse supervision. We believe this integration represents a meaningful contribution to TSC XAI.
>
> >### R5: Limitations
>
> We add a Limitations section discussing aspects, e.g., dependence on the quality/diversity of the expert pool, computational costs of Stage 1, or the gap between model-faithfulness and human semantic interpretability.

---

> > ### Author Rebuttal · Reviewer_ZLM3 · 2026-04-04
> >
> > Thank you for the rebuttal!
> >
> > **Metric-only experiments**: Thanks for the experiments! This clarifies the issue for me.
> >
> > **Need for faster XAI**: ECG monitoring is an interesting example, and really helps ground the advances in the paper. This also highlights the need for closer collaboration with domain experts. In this case, it would be interesting (in future work) to determine whether computational efficiency of explanation computation is really the fundamental bottleneck for the deployment of XAI in clinical settings; and what role it plays in human-in-the-loop diagnosis where humans are often the bottleneck. Regardless, I hope these grounded motivations are added to the introduction of the paper to help connect it to the real world.
> >
> > **Human-in-the-loop evaluation**: Thanks for the pointer about the MIT-ECG dataset having the ground truth expert annotations; that definitely helps.
> >
> > The response has addressed most of my questions, so I will raise my score.
> >
> > **Suggestion for the authors**:  Regardless of the conventions in the TSC XAI community, I urge the authors in the future to consider grounding their work in real downstream use-cases; and perform expert-in-the-loop validation. This is critical for XAI, without which the significance or downstream impact of research will continue to be limited. Please see (https://arxiv.org/abs/2110.10790) for some discussion on this topic.

---

> > > ### Author Response · Authors · 2026-04-04
> > >
> > > We are sincerely grateful to the reviewer for the thoughtful engagement and for raising the score. We appreciate the recognition that our ablation study, ECG monitoring motivation, and expert annotations help address the concerns. We will incorporate all of these into the revised paper.
> > >
> > > We wholeheartedly agree on the importance of closer collaboration with domain experts. The suggested paper "Human-Centered Explainable AI (XAI): From Algorithms to User Experiences" offers valuable guidance that we will study carefully. We plan to pursue expert-in-the-loop validation with real downstream use-cases in future work. As the reviewer rightly emphasizes, bridging the gap between algorithmic XAI and real-life applications is essential for meaningful impact. We are grateful for this direction.

---

### Decision · Program_Chairs · 2026-04-30

**Decision:**

Accept (regular)

**Comment:**

After carefully reading the reviewers’ thorough evaluations and the authors’ diligent responses, I find that although some issues remain, such as the lack of grounded use cases, the reviewers’ main concerns have overall been adequately addressed. Moreover, the paper itself is strong in its motivation for the XAI method, its experimental validation, and its presentation and writing. Therefore, I recommend that this paper be accepted.